# Quantifying uncertainty of high-resolution remotely sensed topographic surveys for ephemeral gully channel monitoring

Robert R. Wells[1], Henrique G. Momm[2], Carlos Castillo[3]

[1]National Sedimentation Laboratory, Agricultural Research Service, United States Department of Agriculture, Oxford, Mississippi, 38655, U.S.A.
[2]Department of Geosciences, Middle Tennessee State University, Murfreesboro, Tennessee, 37132, U.S.A.
[3]Department of Rural Engineering, University of Córdoba, Córdoba, Spain.

*Correspondence to:* Henrique Momm (henrique.momm@mtsu.edu)

**Abstract.** Spatio-temporal measurements of landform evolution provide the basis for process-based theory formulation and validation. Overtime, field measurement of landforms has increased significantly worldwide, driven primarily by the availability of new surveying technologies. However, there is not a standardized and/or coordinated effort within the scientific community to collect morphological data in a dependable and reproducible manner, specifically when performing long-term small-scale process investigation studies. Measurements of the same site using identical methods and equipment, but performed at different time periods may lead to incorrect estimates of landform change as a result of three-dimensional registration errors. This work evaluated measurements of an ephemeral gully channel located in agricultural land using multiple independent survey techniques for locational accuracy and their applicability to generate information for model development/validation. Terrestrial and un-maned aerial vehicle-based photogrammetry platforms were compared to terrestrial LiDAR, defined herein as the reference dataset. Given the small-scale of the measured landform, alignment and ensemble equivalence between data sources were addressed sine qua non through post-processing. Utilization of ground control points were prerequisite to three-dimensional registration between datasets and improved confidence in the morphology information generated. Not one of the methods was without limitation; however, careful attention to project pre-planning and data nature will ultimately guide temporal efficacy and practicality of management decisions.

**Keywords:** close-range photogrammetry, un-manned aerial vehicles, ephemeral gully, soil erosion, remote sensing.

## 1 Introduction

Spatio-temporal measurements of landform evolution provide the basis for process-based theory formulation and validation. Field measurement of landforms has increased significantly worldwide, driven by the availability of new surveying technologies. Recent improvements include, but are not limited to: aerial and terrestrial Light Detection And Ranging (LiDAR) systems (Kukko et al., 2012; Vinci et al., 2015; Eitel et al., 2016; Hawdon et al., 2016), integrated Un-manned Aerial Vehicles (UAV) utilizing both photogrammetric and LiDAR payloads (Bachrach et al., 2012; Bry et al., 2015; Honkavaara et al., 2016), Real Time Kinematics (RTK) (Rietdorf et al., 2006), terrestrial photogrammetric systems (James and Robson, 2014; Gómez-Gutiérrez et al., 2014; Di Stefano et al., 2016; Marzolff, 2016), and low-cost and/or freeware coupled Structure from Motion (SfM) and MultiView-Stereo (MVS) photogrammetric software (Castillo et al., 2012; Castillo et al., 2015; Smith and Vericat, 2015; Piermattei et al., 2016). However, the buyer must beware that these systems can be prone to misinterpretation (Wheaton et al., 2010) and, even the "high-resolution" equipment can provide misleading information (e.g. Vinci et al., 2015; Figure 13b).

Research efforts should focus on a standardized and/or coordinated effort within the scientific community to collect

morphological data in a dependable and reproducible manner, specifically when performing long-term process investigation studies (Castillo et al., 2016).

Ephemeral gullies are often defined as small channels, on the order of a few centimeters in depth, predominantly in agricultural fields (SSSA, 2008). The emergence, evolution and persistence of these concentrated flow path erosion features are controlled by the combined effects of flow, slope, soil properties, topography and vegetation characteristics (Zevenbergen, 1989; Castillo et al., 2016). The term ephemeral refers to the fact that agricultural producers often erase these channels during regular farming operations (Foster, 2005) and/or flow within these channels is cyclical. The combination of a highly dynamic lifespan with the relatively small-scale channel features require high accuracy measurements with high temporal and spatial resolution.

Many studies have been conducted to assess topographic accuracy of ephemeral/classical gully morphological measurements using a wide range of systems (e.g. Casalí, et al., 2006; Gómez-Gutiérrez et al., 2014; Di Stefano et al., 2016). Among them, LiDAR data have been used as the reference for the evaluation of secondary remote sensing systems and physical contact systems. Traditional airborne LiDAR studies have primarily focused on quantifying locational error from datasets generated by air-borne systems, where locational variations are the result of coalesced errors generated by inaccuracies of Global Positional Systems (GPS), aircraft Inertial Measurement Unit (IMU) and overall timing of the system (Hodgson and Bresnahan, 2004). LiDAR positional errors can also be the result of an interaction between the laser pulse and features with sharp relief change or occlusions that result in multiple returns from one laser pulse (Milenković et al., 2015). Evaluations of the accuracy of topographic information using airborne LiDAR are often compared with discrete sample locations and/or man-made targets with known coordinates (Hodgson and Bresnahan, 2004; Csanyi et al, 2005). Despite the large number of studies and methods developed to quantify positional errors in traditional airborne LiDAR surveys, this type of survey does not offer the temporal and spatial resolution necessary for quantitative monitoring of small-scale geomorphological characteristics (i.e. ephemeral gullies) necessary for process description; although, recent developments in UAV LiDAR systems provide 10mm survey-grade accuracy, one million measurements per second, and 360º Field of View (FoV), all in <1.6 kg payloads. These UAV LiDAR systems can range from $100k to $400k (US dollar), dependent upon level of accuracy and data collection rate (see http://www.rieglusa.com as an example).

At a finer scale, investigation of ground-based/terrestrial LiDAR has demonstrated a high locational accuracy (~2mm) and noted the importance of appropriate spatial sampling density for ephemeral and classical gully investigation (Momm et al., 2013a). Topographic representation of gully channels requires datasets with a specific minimum sampling density, which is dependent on site-specific topographic characteristics (Castillo et al., 2012; Momm et al., 2013a). Overlapping the same area by multiple scans increases the overall sampling density and assists in occlusion and shadow avoidance, while normalizing spatial resolution.

Studies involving various surveying techniques of concentrated flow paths revealed a wide range of quality, accuracy, cost, and field campaign effort (Momm et al., 2011, 2013b; Castillo et al., 2012; Wells et al., 2016). Among the surveying techniques considered, photogrammetry has been shown to provide simple but robust measurement of small-scale changes in geomorphologic characteristics within agricultural fields (Castillo et al., 2012; Gesch et al., 2015; Wells et al., 2016). Further, a wide variety of platforms and techniques have been used to capture images, including kites (Marzolff, et al., 2003), backpacks (Wells et al., 2016) and UAVs (Ries and Marzolff, 2003; Bachrach et al., 2012; James and Robson, 2014; Cook, 2017). Erosion monitoring programs based on photogrammetry have several advantages when compared to other surveying techniques. Photogrammetric field surveys do not interfere with farming operations, as it is non-obstructive; field campaigns are extremely efficient; and often they do not require specialized technical skillsets to implement (James and Robson, 2012). However, photogrammetric results can vary as a function of controlling parameters used during data collection and processing (Eltner et al., 2016).

A particular point of interest is the general query posed by Wheaton et al. (2010) concerning real geomorphic change. With these evolving technologies, our ability to collect topographic information is seemingly limitless. At what point can we agree that the results describe "real" change over noise? The alignment of temporal-topographic elements is the most critical step when planning small-scale erosion studies (Smith and Vericat, 2015). Reliance on control points is the foundation of classical surveying. All surveys must close with a shot back to the initial occupation point. This too is the initiation of error propagation. A multitude of solutions exist for each set of photos and/or LiDAR points; however, the unique solution is bounded by the spatial and vertical positioning of the control points (Micheletti et al., 2015). Provided that alignment can be controlled, the next operation typically involves a culling process of some sort as the data shift into organized units.

The conversion of irregularly sampled point clouds into regular grids, referred to as Digital Elevation Models (DEMs), is extremely common as most flow routing algorithms and Geographic Information Systems (GIS)-based soil erosion modeling technology are designed to work using these digital representations. As a result, a large number of studies have been conducted to evaluate DEM representation as affected by sampling intervals and interpolation algorithms (Aguilar et al., 2005; Ziadat, 2007; Bater and Coops, 2009; James and Robson, 2012; 2014), and by DEM spatial resolution (Zhang and Montgomery, 1994; Kienzle, 2004; Momm et al., 2013a).

The majority of previous studies have focused on accuracy evaluation of a specific photogrammetric survey method at a single time period. Varying sensors, platforms, and processing methods can yield different results (variations in sampling densities, gaps and noise). Furthermore, measurements of the same site using identical methods and equipment, but performed at different time periods can also lead to three-dimensional registration errors. Therefore, the scope of this work was to evaluate multiple survey techniques and provide a framework for temporal studies of ephemeral gully channels. Three surveying platforms, with varying parameters, were independently evaluated for locational accuracy and applicability in generating information for model development/validation. The objectives of this study are twofold: to quantify the overall accuracy of the different survey configurations and to develop practical guidelines for the design and implementation of future ephemeral gully monitoring studies.

## 2 Methods

### 2.1 Study Site

The study site was located in the northwest corner of Webster County, Iowa, U.S.A. (Figure 1). Farming is the dominant enterprise in Webster County. The crop rotation was a corn-soybean rotation. Total annual precipitation is about $873\ mm$, of which, 70 percent usually falls between April and September. The Area Of Interest (AOI) within the field survey was a small reach ($1.9\ m\ x\ 1.3\ m$; $2.47\ m^2$) of a $150\ m$ long ephemeral gully, oriented south to northwest, eroding Clarion loam (fine-loamy, mixed, superactive, mesic Typic Hapludolls) at the upper (south) extent, Terril loam (fine-loamy, mixed, superactive, mesic Cumulic Hapludolls) on the intermediate slopes, and Webster clay loam (fine-loamy, mixed, superactive, mesic Typic Endoquolls) within the lower relief section of the field (Figure 2). Within the AOI, the soil was Clarion loam (Figure 2B).

### 2.2 Field Survey

Field surveys were conducted using three independent modes: ground-based/terrestrial LiDAR, ground-based/terrestrial photogrammetry, and airborne photogrammetry. The surveys yielded twelve datasets (Table 1; dashed lines represent delineation between survey modes). The terrestrial LiDAR was considered the reference dataset, due to perceived superior accuracy. All surveys were run independent of each other and completed on the same day. Each dataset was represented using the NAD83 UTM15N coordinate system.

In this study, the terrestrial LiDAR point cloud was generated using TopCon ScanMaster software (https://www.topconpositioning.com/software/mass-data-collection/scanmaster), all terrestrial photogrammetric point clouds were generated using PhotoModeler Scanner software (www.photomodeler.com/products/scanner/default.html) and all airborne photogrammetric point clouds were generated using Pix4Dmapper Pro software (https://pix4d.com/pix4dmapper-pro/). It is acknowledged that the selection of input parameters influences the sampling intensity and local elevation variance; however, the quantification of the influence of input parameters on the output is beyond the scope of this study. Here, similar survey methods used the same input parameters to generate point clouds.

### 2.2.1    Differential Global Positioning System (dGPS) of Ground Control Points (GCPs)

Site preparation began by locating a state monument point (Figure 2A) and laying out 406 x 406 mm quad-triangle, white-on-black sheet GCPs, considering the long and short axes of the field as well as the high and low elevations within the field boundary, herein considered to be the field GCPs (10 total). One additional set of GCPs with rad coded targets was arranged along the gully channel (four pins at the location; Wells et al., 2016), herein considered to be the channel GCPs (4 total). All GCPs were surveyed using TopCon GR-3 dGPS survey equipment (TopCon Corporation, Tokyo, Japan; 10 $mm$ horizontal and 15 $mm$ vertical kinematic accuracy) to obtain relative position in reference to the state monument point. A static occupation (6 $hrs$; 3 $mm$ horizontal and 5 $mm$ vertical accuracy) was initiated with the base station, then all GCPs (field, channel and state monument) were surveyed with the rover (20 $sec$ collection interval). All survey data was corrected using an OPUS (National Geodetic Survey) solution of the base station location, processed with reference to the state monument point (6 $mm$ overall vertical accuracy). Both (field and channel) GCP positions were used to adjust point clouds from the LiDAR and photogrammetric surveys of the site.

### 2.2.2    Terrestrial LiDAR Survey

The terrestrial LiDAR survey was conducted using a TopCon GLS 1500 (TopCon Corporation, Tokyo, Japan; 4 $mm$ single point and 2 $mm$ surface accuracy with a spot size $< 6\ mm$). The system operates in similar fashion to standard total stations. For each laser pulse, the system records X, Y and Z coordinate values with respect to the position of the scanner sensor (local coordinate system), the intensity of the returned signal (reflectance) and spectral information from an integrated digital camera within the instrument. Local coordinates are transformed into global coordinates during post-processing by entering the external geometry coordinates (i.e. absolute position determined from Post-Processed Kinematic survey) of GCPs. The AOI (demarked by channel GCPs; 2.47 $m^2$; Figure 2B) within the field boundaries, covering the gully channel, was surveyed with one scan resulting in a total of 5,613,334 scan points.

Given the level of user control of input parameters and the high locational accuracy of terrestrial LiDAR systems, this survey method was selected as the reference to which all other survey methods were compared. However, it is important to acknowledge that this survey method does have limitations.

In surveys with high sampling intensity, it is common for the same location on the ground to be hit by multiple laser pulses. This yields datasets with high sampling intensity but with a range of elevation values for the same location (i.e. fluff) given the vertical accuracy of the system. In this study, this elevation variability is estimated to be approximately $\pm 2\ mm$. The sensor operates in the near-infrared portion of the electromagnetic spectrum (1,535 $\eta m$) and in this spectral range; electromagnetic energy is absorbed by water (Aronoff, 2005). In locations with water features, the sensor emits laser pulses but no laser pulse is reflected back to the sensor, preventing the range calculation for that particular pulse. During data collection for this survey, a shallow film of water was present in the gully channel (Figure 3A) and, as result, no points were recorded in the water-covered region (Figure 3B) (e.g.

Gómez-Gutiérrez et al., 2014). Sampling gaps in LiDAR surveys can also be attributed to vertical features that limit the sensor line-of-site, referred to as shadowing (Figure 3C). The basic principle of LiDAR technology is to measure the time needed for an individual laser pulse to travel from the transmitter to the target and back to the receiver allowing the range distance to be calculated (Wehr and Lohr, 1999). However, in certain situations, as the scanner moves along the scan arc, the laser footprint hits an area just past the edge of a surface where the next return appears to be from a distance greater than expected (i.e. occlusion). In this case, the gap is linearly filled by equally spaced points (Figure 3C highlight). A shadow of the obstruction appears in the dataset. These artificial points may be filtered by intensity and multiple scan positions may be used to discriminate the features. Here, the AOI was slightly decreased in areal size to omit GCP occlusions from the dataset.

### 2.2.3    Terrestrial Photogrammetric Survey

Terrestrial photogrammetry was conducted using a Nikon D7000 16.2 MP camera (Nikon Inc., Melville, NY) with calibrated 20mm lens (Gesch et al., 2015; Wells et al., 2016). The camera was mounted to a backpack frame connected to an iPad mini (Apple, Cupertino, CA) through a WiFi CamRanger hub (Camranger LLC, http://www.camranger.com) (Wells et al., 2016). Multiple images were collected around the channel GCPs, including views from each corner and all sides. Still images captured by the camera were transformed to point clouds using PhotoModeler Scanner photogrammetric software. Initial data processing included: aerial triangulation and bundle adjustment, camera position and orientation. Following initial processing, channel GCP positions (i.e. global external geometry) were included to optimize point cloud accuracy.

### 2.2.4    UAV-based Photogrammetric Survey

Two UAV platforms were used to collect airborne photography (https://www.sensefly.com/home.html): fixed-wing (eBee) and quadrotor (albris). The fixed-wing platform had a 12 MP nadir camera (i.e. belly mount; Canon S110 RGB) and was deployed (eMotion2 v2.4.10) to capture the entire field boundary (Figure 2A) by throwing the craft in the air, then the craft flies, captures images and lands itself. Deployment software parameters were: Altitude (117 m, 58 m); Resolution ($41\ mm$, $20\ mm$); Latitude Overlap (80%, 50%), Longitude Overlap (80%, 50%), Image Collection (356, 569), and Image Format (CR2(RAW)) for the two respective flights. The quadrotor platform had a 38 MP camera mounted within an 180$^{\circ}$ vertical range head and was deployed (eMotionX v1.3.0) to capture both, the extent of the gully within the field and specific points of interest (i.e. AOI; Gesch et al. 2015; Wells et al. 2016). The quadrotor was deployed through mission planning software. The craft takes off, flies and captures images, then lands itself.  Deployment software parameters were: Altitude ($35\ m$, $20\ m$); Resolution ($7\ mm$, $5\ mm$), Latitude Overlap (75%, 75%), Longitude Overlap (80%, 80%), Image Collection (96, 146), and Image Format (dng(RAW)) for the two respective flights. During the flights, winds from the southeast ranged from 7 to 10 $m/s$ and skies were clear.

Still images captured by the UAVs were transformed to point clouds using Pix4DMapper Pro photogrammetric software. Initial data processing included: camera calibration, aerial triangulation and bundle adjustment, camera position and orientation. Following initial processing, field GCP positions (i.e. global external geometry) were included to optimize point cloud accuracy. Both, fixed-wing and quadrotor, UAV systems were deployed with a fixed path and common photograph overlap percentage. All missions/deployments were preplanned using flight planning and control software provided by the manufacturer. A mission block and specific area or point of interest were selected, including preferred ground resolution, camera head angle (quadrotor only) and flight altitude, then flight lines for aerial coverage, circular paths with horizontal plane around objects of interest (quadrotor only), image capture points and waypoints were generated prior to deployment. Key flight parameters were displayed in real-time, along with battery level and image acquisition progress, while the autopilot continuously analyzed onboard control data to optimize the flight.

## 2.3 Dataset Alignment

To ensure the highest three-dimensional alignment among all point clouds and consistent spatial coverage by all methods, a three-step preprocessing approach was developed. First, the set of four channel GCPs (white square targets in Figure 2B) were used to generate a rectangular polygon to subset the point clouds in all surveys, assuring that all surveys cover exactly the same ground position. Second, a smaller polygon subset was performed by generating a polygon with a $50\ mm$ reduction on all sides. This was performed to exclude areas close to the channel GCPs and assure the elimination of shadowing and occlusion created by the slightly elevated channel GCPs. Third, the sampling void within the LiDAR dataset, created from the presence of a thin film of water within the channel (Figure 3A), was manually digitized into another polygon and used to remove points from all photogrammetric-generated datasets to assure uniformity between all datasets. Essentially, instead of using interpolated data within this void in the reference dataset, we simply placed a void in all datasets; therefore, we do not introduce bias into the calculations with regard to the water film void within the LiDAR data (e.g. Gómez-Gutiérrez et al., 2014).

Subsequently, since each surveying method was performed using the same set of field and channel GCPs, manual inspection of measured points located coincident with channel GCPs (white square targets in Figure 2B) were used to generate planes (10), one for each dataset with the exception of the Fixed_61m and Fixed_122m datasets, in which no points were located on top of the channel GCPs. The four GPS surveyed coordinates of the center location of the channel GCPs were used to fit a reference plane to be matched by all surveys (black squares in Figure 4). Three-dimensional locational differences between the reference plane generated using the GPS survey (black squares in Figure 4) and the planes of each surveyed dataset (LiDAR and photogrammetry) were calculated using the Iterative Closest Point (ICP) algorithm (Besl and Mckey, 1994; James and Robson, 2012; Micheletti et al., 2015) implemented in MATLAB (MathWorks Inc., Natick, Massachusetts) and, since no scale issues were observed, no scaling factor was implemented in the ICP. The ICP algorithm minimizes the locational differences between two sets of three-dimensional point clouds and outputs a 3x3 rotation angle matrix, $R$, and a 1x3 translation vector, $T$ (Eq. 1). These matrices were used to three-dimensionally transform, through rotation and translation, the measured point clouds to best match the reference plane.

$$\begin{bmatrix} X \\ Y \\ Z \end{bmatrix}_{ref} = T + R_{\omega\varphi\kappa} * \begin{bmatrix} X \\ Y \\ Z \end{bmatrix}_{meas} \tag{1}$$

## 2.4 Error Metrics

One of the problems in looking at the data in the original point cloud format was the large difference between the total number of points within datasets (i.e. hundreds to millions), which tends to bias the results; therefore, in the sections that follow, the investigation of point cloud data is complimented by an analysis of gridded data. In the point cloud analysis, each point within the photogrammetry surveys was compared to that within the LiDAR survey. The analysis is carried out in two ways: point normal to the plane (each photogrammetry point was projected normal to a fitted plane of LiDAR points at the nearby position) and spot elevation to Triangular Irregular Network (TIN) (each photogrammetry point was projected up or down to intersect the TIN surface of the LiDAR points). In the gridded data analysis, volume difference and cross-section assessment are performed. Furthermore, this type of data structure (i.e. raster grid) is a common format used to estimate soil loss volumes and generate cross-sections for modeling exercises (Dabney et al., 2014). And, the gridded data introduce a common means of discussing differences between the survey methods.

### 2.4.1 Sampling Intensity, Local Variance, and Spatial Pattern

Sampling intensity is defined as the number of points per unit of area. Investigation of the sampling intensity spatial variation can reveal over/under sampled locations. Under sampled locations may be potential sources of error in quantifying geomorphologic change (i.e. cross-sectional areas, volumes, etc.), especially in surfaces with high relief. Sampling intensity was evaluated using the quadrant method (Dodd, 2011), where a virtual regular grid of 1 cm was imposed on each dataset and the number of LiDAR and photogrammetry points falling within each grid was counted and recorded. Similarly, the local elevation variance was evaluated by calculating the elevation range (difference between maximum and minimum) within each grid. The local elevation variance is a function of the terrain characteristics, sampling intensity, survey method, and post-processing parameters.

Two metrics were used to quantify the spatial pattern distribution: distances between events and between events and random points not in the pattern (void space). The G-Function, $G(r)$, defined as the cumulative frequency distribution of nearest-neighbor distances (Lloyd, 2010), provides the conditional probability that the distance between points (event-event) is less than the point distance threshold ($r$). The empirical distribution is obtained for each distance $r$ by counting the number of points at distances less than or equal to $r$ from each point within the AOI. The theoretical distribution is obtained by assuming a completely random pattern with density ($\lambda$; estimated by the ratio of total number of points, divided by the area of the AOI), modeled as a Poisson process. Empirical values closer to the theoretical values indicate a random distribution, empirical values above the theoretical values indicate clustering, and empirical values below the theoretical values indicate a more regular distribution (Bivand et al., 2008). The F-Function, $F(r)$, defined as the cumulative frequency distribution of the distance to the nearest point in the AOI from random locations not represented within the AOI (Lloyd, 2010), provides the probability of observing at least one point (event) closer than $r$ to an arbitrary point within the AOI ("empty space" or "void" distances). Estimated and theoretical distributions are obtained similar to that of the G-function. The interpretation of graphed observed versus theoretical values indicates a regular pattern when the observed is above the theoretical values and clustering when it is below (Bivant et al., 2008). These point pattern analyses were performed using the Spatstat package in the R software package (Baddeley and Turner, 2005; Bivand et al., 2008).

Furthermore, points with the same X, Y and Z to the fifth decimal place were removed from the LiDAR dataset, indicating collection of redundancy information.

### 2.4.2 Vertical and Horizontal Displacement using Point Tangential Projection to Plane

Vertical and horizontal displacement between the LiDAR and photogrammetry measurements in each dataset were quantified using the normal projection of each photogrammetry point into a plane fitted to the nearest LiDAR points (Figure 5A). For each photogrammetry point (green circle in Figure 5A), the nearest (within 25 $mm$ sphere) LiDAR points were selected (black dots in Figure 5A), a plane was fitted to the selected LiDAR point cloud points, the photogrammetry point was normally projected onto the plane, the coordinates of the intersection point (red circle in Figure 5A) recorded and statistics were generated. This analysis was performed for all datasets using an in-house developed Python script, where

$$D_x = \left( Z_x^{plane} - Z_x^{photo} \right)$$
$$D_y = \left( Z_y^{plane} - Z_y^{photo} \right) \tag{2}$$
$$D_z = \left( Z_z^{plane} - Z_z^{photo} \right)$$

### 2.4.3 Vertical Displacement using Spot Elevation

The three-dimensional point cloud representing the reference dataset (LiDAR) was converted into a TIN (Figure 5B). Each point in the photogrammetry dataset (green circle in Figure 5B) was compared to the LiDAR TIN by fixing the photogrammetry X and

Y coordinates while varying the Z-coordinate up or down until the point intersected the TIN (red circle in Figure 5B). The Z-coordinate at intersection was recorded. This analysis was performed using ArcGIS (Esri, 2011).

$$D_z = \left( Z_z^{TIN} - Z_z^{photo} \right) \tag{3}$$

The vertical displacement at a location is calculated by subtracting the photogrammetry elevation from the LiDAR elevation of the
TIN. Descriptive elevation statistics were generated based on all raster grid cells.

### 2.4.4   Gridded Surface Assessment

All point clouds were converted into 5x5 mm regular raster grids using linear interpolation. The two upstream channel GCPs served as the base of the rectangular grid; roughness of individual datasets was highly dependent upon sampling intensity. Volume difference calculations were performed between the LiDAR raster grid and the photogrammetry raster grid. Two metrics were
calculated, volume difference and absolute volume difference (Eqs. 4 & 5).

$$V_{diff} = \sum_{i=1}^{n} \left[ \left( Z_i^{lidar} - Z_i^{photo} \right) * ca \right] \tag{4}$$

$$V_{acc} = \sum_{i=1}^{n} \left[ \left| Z_i^{lidar} - Z_i^{photo} \right| * ca \right] \tag{5}$$

where $Z^{lidar}$ is the reference elevation, $Z^{photo}$ is photogrammetry elevation, $ca$ is the raster grid cell area (0.000025 $m^2$), and $n$ is the total number of raster grid cells.

### 2.4.5   Gridded Cross-Section Assessment

Gully modeling technologies often use cross-sections as basic modeling units. With the objective to assess the error introduced by each survey to cross-section analysis, the raster grid surfaces were used to generate nine (9) cross-sections. For each cross section, various assessments were conducted, including: minimum elevation, maximum elevation, mean elevation, variance, linear modeling ($R^2, p, SEM$), and area calculations. Area above the curve was selected as one of the metrics to quantify cross-section
accuracy, given as

$$A_c = \sum_{i=1}^{n} \left[ (353.00 - Z_i) * \Delta_{dist} \right] \tag{6}$$

where $A_c$ is the area for cross-section $c$ in square meters, $Z_i$ is the elevation at point $i$ in the cross-section, and $\Delta_{dist}$ is the distance between points in the cross-section (0.005 $m$). An elevation constant (353.00) was used to adjust all cross-section elevations, due to local elevation relation to mean sea level, thereby truncating area values. The deviation of the area estimates from the LiDAR
were calculated using:

$$A_{dev} = \left( \frac{A_{obs} - A_{est}}{A_{obs}} \right) * 100 \tag{7}$$

### 2.5  Dataset Scoring

Since there is no true standard of judging the performance of the measurements provided herein, a system of scoring was developed
to grade the photogrammetry data with regard to the LiDAR data (Table 2). Scores between 1 and 11 were assigned to each evaluation category using both, point cloud and gridded data. For example, the difference in absolute volume was assigned decreasing scores (1→11) for increasing volume difference and, correlation coefficients were assigned decreasing scores (1→11) for decreasing correlation. Simply, if a variable had a positive impact, it received a higher score and all variables were equally weighted. Each score is defined in Table 2.

# 3 Results

## 3.1 Sampling Intensity and Point Pattern Evaluations

The point clouds evaluated here had a large variability in sampling intensity (from 1 to $> 250$ points per $cm^2$; Figure 6). The difference in point sampling influences micro-topography and apparent roughness, and may lead to bias in volume estimation. If the surface is rough, the effect will be greater (Figure 7). Point counts are very low for the fixed-wing flights in comparison to the other methods and the sparse point count leads to interpolation (filling) during raster gridding (Figure 6), while elevation range is very similar for all methods with the exception of the fixed-wing datasets (Figure 7).

Point pattern analysis was examined using the G- and F-functions (Figure 8). Each tests through an assumption of completely spatially random (homogeneous Poisson process), although interpretations for clustering and regularity are in opposition for each test (i.e. regular point spacing outcome for the G-function is below gray confidence bounds and for the F-function it is above). At first, the confidence bounds (gray envelope bounding the theoretical values (red dash line)) show that the Fixed_122 has a sparse point count. Looking at the G-function results, the data are clustered at distances of $0.02\ m$ (Fixed_122), $0.002\ m$ (Quad_20) and $0.001\ m$ (Grd_8A), then regularly distributed. This indicates that small distances occur less often than expected under the assumption of spatial randomness. The F-function results say that the data are randomly distributed to $0.06\ m$ (Fixed_122) and $0.025\ m$ (Quad_20), and clustered for Grd_8A (i.e. on short distances, fewer points are encountered than for a random pattern); although, the scale of $r$ should be acknowledged ($< 3\ mm$). All twelve datasets yielded observed G-function values below the theoretical values, indicating a regular sampling pattern. The terrestrial photogrammetric surveys did show slight clustering at small distances ($<\ 3mm$). Therefore, based on these metrics, a regular sampling pattern was observed, indicating that all locations within the study area were sampled with a similar sampling pattern (no areas where over sampled nor under-sampled) for all twelve datasets in this study.

## 3.2 Vertical and Horizontal Displacement Evaluations

Graphical representation of both spot elevation to TIN and normal to fitted plane for the fixed-wing flight at $122\ m$ altitude (Fixed_122 (Figure 9)), and the four photo-pair (Ground_8A (Figure 10)) are provided for comparative purposes. In the Fixed_122 spot and normal analysis (Figure 9), the range was larger for the spot ($\pm 0.06\ m$) than normal ($\pm 0.02\ m$) and residuals suggest a non-linear response, potentially attributed to over-smoothing of the surface and lack of pre-processing (slope of blue line). Residuals for the Ground_8A (Figure 10) have a constant variance and do not show either X- or Y-axis bias. Clearly, outliers can be identified (Figure 10; $\sim 0.01\ m$) and it seems as though the spot analysis provides a larger variance (i.e. amplified residual signature) than the normal analysis, which may be attributed to the 2.5 $cm$ sphere used to define the plane in the normal analysis (i.e. smoothing).

Very similar results were obtained with these two methods. All datasets had negligible mean displacement in the X and Y-directions (Table 3). The standard error of Z in the normal to plane analysis (Table 3) ranged from 0.3 $mm$ (Ground_8A, Ground_6A, Ground_4B, Ground_4C, Ground_2B) to 2.9 $mm$ (Fixed_122). The mean displacement for the normal to plane analysis in the Z-direction (Table 3) ranged from 0.2 $mm$ (Quad_35) to 7.4 $mm$ (Fixed_61). For the vertical spot to TIN analysis (Table 4), standard error ranged from 0.4 $mm$ (Ground_8A, Ground_6A, Ground_4B, Ground_2B) to 2.1 $mm$ (Fixed_122), and mean displacement ranged from 0.1 $mm$ (Ground_8A) to 35 $mm$ (Fixed_122). Here, with the spot to TIN analysis, also note that the LiDAR point cloud was evaluated resulting in a standard error of 0.3 $mm$ and a mean displacement of 0.1 $mm$; both results are similar to Ground_6A and Ground_8A for standard error and Quad_35 and Ground_4A for mean displacement. The mean elevation difference between photogrammetry and LiDAR was approximately 5.3 $mm$, between the quadrotor and LiDAR was 3 $mm$ and

between the fixed-wing and LiDAR was 25 $mm$. Perhaps the ten-fold increase derived from the fixed-wing flights was simply because they were not pre-processed (affine transformation was not applied); however, this result does show the importance of common reference points between surveys.

Most of what is reported here is due to point cloud alignment during the preprocessing step discussed earlier (i.e. GCP alignment for each, except fixed-wing flights). The mean elevation difference was approximately 5$mm$ for all datasets (Tables 3 and 4). Positive values indicate that the LiDAR data was, on average, higher than the photogrammetry data and negative values indicate that the LiDAR data was, on average, lower than the photogrammetry data. Similarly, when contrasting the photogrammetry elevation with the elevation of the normal point through linear regression (Table 3), the slope of the fitted line is very close to unity for all methods except Fixed_122, indicating spatially variable discrepancies (higher elevation differences at one region than the rest of the study site).

### 3.3 Gridded Surface Evaluations

The conversion of point clouds with irregularly spaced points, and spatially varying sampling intensity point clouds, into regular raster grids affected each dataset differently (Figure 11). For example, the LiDAR dataset contained a high sampling intensity (> 100 points per cm$^2$) with relatively large elevation variability of the points within a raster grid cell. Therefore, the interpolation procedure generated a significantly smoothed surface (e.g. Eitel et al., 2011). Conversely, the fixed–wing surveys had a low sampling intensity and the interpolation procedure linearly filled the gaps, potentially generating a significantly smoothed surface that differs from the "natural" surface. Variance shows roughness is similar for most of the surveys, with the exception of Fixed_122 where the variance is extremely low (0.0005; Table 5). This result merely points out that the Fixed_122 is extremely smooth in comparison to the other surveys, due to the low sampling density (Figure 6) and enhanced interpolation between points for the high spatial resolution raster grid (0.005 $m$ cell size).

One of the most important measurements for gully monitoring is the volume difference between surfaces (Table 6). Given the small scale of this type of erosional feature (on the order of few cm), it is vital to have a good understanding of the expected error for each method. Among similar collection methods (terrestrial photogrammetry), the absolute volume difference (Table 6) ranged from 0 to 52%, in comparison to Ground_8A; although these differences were extremely small in reality (i.e. range of 0.0062 to 0.0105 $m^3$). Performance ranking, in terms of absolute volume difference, was Quad_35, Ground_8A, Ground_4A, Ground_6A, Quad_20, Ground_4C, Ground_4B, Ground_2B, Ground_2A, Fixed_61 and Fixed_122 (10 to 164% absolute volume difference for Ground_8A and Fixed_122, respectively, in comparison to Quad_35; Table 6). Variance in elevation difference between the LiDAR and photogrammetry data were all quite similar (Table 5), with the exception of the fixed-winged flights (effect of interpolation). In terms of the elevation range of the data (Table 5), the terrestrial photogrammetry and quadrotor flights were within 1.15% of the LiDAR range and appeared to be very similar (Figure 6), with the exceptions of Ground_2A (too rough; 8.2 % roughness increase) and Fixed_122 (too smooth; 84 % roughness decrease).

### 3.4 Gridded Cross-section Evaluations

In the gridded elevation evaluations (Min, Max, Mean; Table 5; Figure 12), absolute difference from LiDAR was less than 0.02%, and these differences were only seen in the fixed-wing flights. The variance (i.e. roughness), however, does shoe that absolute differences from LiDAR were 131% (Fixed_122), 23% (Fixed_61), and 9% (Grround_4A). Comparing elevation information (Table 7; Figure 12) between photogrammetric cross-sections and LiDAR cross-sections through linear regression, indicates a coefficient of determination larger than 0.98 for all datasets, excluding the two fixed-wing flights. The standard error of this regression was less than 10 $mm$ for Quad_20, Quad_35, Ground_2B, Ground_4B, Ground_4C, Ground_6A, and Ground_8A.

Following that, Ground_2A and Ground_4A had a standard error of approximately 17 $mm$ and the two fixed wings had a standard error of 25 $mm$. The average area percent difference for all cross-sections were within 1.5%, while the two fixed wings had 3% (Fixed_61) and 8% (Fixed_122). It is important to mention that the range of area percent difference is within ± 2%, while the fixed-wing systems had up to 15% percent difference. The error is huge for instance if this dataset was intended to be used for the purpose of development/calibration/validation of a soil erosion model.

## 3.5 Dataset Scoring Evaluations

Gridded data performance was led by Ground_8A and Quad_35. Combined category score points ranged from 49 (Ground_8A) to 6 (Fixed_122), terrestrial photogrammetry ranged from 49 to 20, quadrotor ranged from 42 to 35, and the fixed-wing ranged from 9 to 6 (Table 8). For the point cloud analysis, scoring results were, for the most part, very similar. The terrestrial photogrammetry surveys all score very high, with the quadrotor falling in the middle and the fixed-wing at the bottom. Overall, scoring ranged from 140 to 24, with the terrestrial photogrammetry leading the group. As the number of photos increased, so too did the sample density; however, the four photo-pair (Ground_8A) was less dense than either the six photo-pair (Ground_6A) or two photo-pair (Ground_4B, Ground_4C), which may be associated with higher accuracy in pixel matching or the addition of inferior image/s to the project. However, it is noteworthy to add that sampling intensity did increase as the UAV altitude decreased, although the Quad_35 did outperform the Quad_20 in a number of categories.

## 4  Discussion

### 4.1  Methods Comparison

Two photogrammetric software packages (Pix4DMapper Pro and PhotoModeler Scanner) were used to generate solutions for the UAV platform and terrestrial photogrammetry surveys. Pix4DMapper Pro uses a larger number (>=3) of overlapping photos, while PhotoModeler Scanner can offer solutions with only two overlapping photos. These software packages differ in the level of user control options for processing and point cloud generation and, point clouds processed by different software packages and/or users could yield very different solutions; although this aspect was not investigated here.

An alarming concern in this analysis was the realization that rotation and translation (Figure 4) was required to ensure that all data was properly aligned. The LiDAR global coordinates were the same as those used for the fixed-wing and quadrotor flights (i.e. field GCPs) and, the channel GCPs were also utilized to optimize the LiDAR point cloud solution. In all terrestrial photogrammetry point cloud solutions, the same set of global coordinates (channel GCPs) were used. One might expect that the solutions would converge without the need to manipulate the point clouds in post-processing; however, not one of the solutions contained the exact positions of the channel GCPs, including solutions generated using the same platform but with varying processing parameters. For example, three-dimensional registration discrepancies were detected between LiDAR solutions and solutions from the quadrotor platform at 20 and 35 meters, the fixed-wing platform at 61 and 122 meters, and the terrestrial photogrammetry surveys. This realization does offer extreme difficulty in temporal studies of ephemeral erosion processes, no matter the choice of resolution, platform and/or processing parameters.

Initially, an attempt was made to analyze all datasets in their original form; however, two limitations to the approach were noted: the lack of three-dimensional registration between the datasets skewed efforts to quantify individual point accuracy and, more importantly, reduced confidence in the geomorphology information generated; and the difference in point sampling density, ranging from hundreds (fixed-wing platform) to millions (LiDAR), biased results. Therefore, the discussion presented herein relates to solutions that have been altered from the original solutions produced by the respective software packages. Comparison

of measured F and G functions with estimated theoretical spatial distribution under the Complete Spatial Randomness assumption suggested that all datasets did not present any spatial clustering, therefore indicating that the study site was sampled uniformly (regular spatially distributed data throughout the study site) (Figure 8). The main difference between datasets were the scale, in which terrestrial photogrammetry and quadrotor airborne photogrammetry yielded sub-centimeter distances as a result of the large number of samples when compared to the fixed-wing airborne photogrammetry. Results from the point evaluations suggest that the Fixed_122 data is clustered below $r = 20\ mm$; however, the very same point cloud was interpreted to a $5\ mm$ raster, so part or all of the metrics associated with these flights may be biased.

The normal projection and vertical spot analysis place the mean elevation for quadrotor flights at $2.9\ mm$ below the LiDAR, for terrestrial photogrammetry at $5.0\ mm$ below the LiDAR and for the fixed-wing flights at $16\ mm$ above the LiDAR. The range of cross-sectional elevation from all terrestrial photogrammetry was within 1% of the LiDAR and, if we drop the rough sample (Ground_2A), the difference falls below 0.2% (e.g. Gómez-Gutiérrez et al., 2014; Di Stefano et al., 2016). All seemingly acceptable for terrain mapping and, perhaps even process development; however, if it is assumed that the bulk density of the soil is $1500\ kg/m^3$, then the soil mass difference for the quadrotor flights is $5.6\ kg$ (erosion/elevation depletion mass), for terrestrial photogrammetry is $9.8\ kg$ (erosion/elevation depletion mass), and for the fixed-wing flights is $44.8\ kg$ (deposition/elevation enhancement mass). Furthermore, if the size of the study area ($2.47\ m^2$) is projected onto a $1\ ha$ field, the elevation distortion is anywhere from 23,000 to $181,000\ kg$ of material. Substantial. Another way to visualize this data would be to look at the area calculations, from which there is 1.06% decrease in cross-sectional area (Figure 10) for Ground_8A over that of the LiDAR. One percent is a very low difference and amounts to an impacted area of $0.03\ m^2$. Again, when considering the site projected to the $1\ ha$ field, the impacted area is on the order of fifty times the original measurement area ($121.5\ m^2$). These findings are reflective of the decision to keep all collected data and further promote the importance of data uncertainty analysis (Wheaton et al., 2010).

Another interesting finding was the difference between solutions from terrestrial photogrammetry (varying number and/or orientation of photo pairs). Solutions from Ground_2A and Ground_2B datasets both used only one photo pair, however, results from the analysis indicate a superior solution generated from the Ground_2B pairing (i.e. upstream/downstream orientation; Table 1). This could be potentially attributed to the orientation of the images in relation to the channel, in which differences in illumination could hamper the photogrammetric process of automated pixel matching between each photo pair (Marzoff and Poesen, 2009). Additionally, increasing the number of photo pairs used in the solution, does seem to yield improved solutions. Results from the volumetric analysis show that the Quad_35 was a very close approximation ($0.0002\ m^3$) to the LiDAR and Ground_4A (i.e channel left and right with corner left and right photo pair; Table 1) was within $0.0001\ m^3$ of the Quad_35. Solutions obtained with "corner left and right photo pair" tended to improve estimates. However, within the overall assessment of data performance, the Ground_4A data finished 8[th] and small differences between Ground_6A and Ground_8A suggest a potential threshold in the number of photo pairs to which including additional photo pairs adds marginally to the final quality of the solution. Whether or not the datasets were adjusted spatially in accordance to the channel GCP positions, absolute volume differences were similarly ranked between photogrammetry datasets. Point clouds built from higher photo pairs and flights of lower altitudes produced better results, when compared to terrestrial LiDAR.

## 4.2 Monitoring Guidelines

Long-term photogrammetric monitoring of ephemeral gullies should be performed with systems and procedures that: (1) Provide a minimum sampling density to capture the overall and local terrain characteristics based upon study objectives (i.e. temporal headcut migration process understanding may require sub-centimeter resolution data, while temporal channel meander process understanding may only require decimeter resolution) (James and Robson, 2012; Gómez-Gutiérrez et al., 2014). The planning

phase of the project must consider the physical characteristics of the process to be investigated, study site physical and environmental variables, and the available hardware and software. (2) Utilize static ground control points visible in comparable photo pairs in all time-step surveys (i.e. fixed known points within the scene provide checks to assure proper three-dimensional registration of temporal data) (e.g. Smith and Vericat, 2015). An organized scheme for control points must be realized for detailed multi-temporal quantitative assessment. Small variations in alignment within temporal surveys will introduce error into length, width, cross-sectional area and volume estimates (e.g. Casalí et al., 2015). Repeated realizations of GCP coordinates will always reduce error in survey solutions. (3) Collect the same number of photo pairs using the same sensor and with the same orientation in all time-step surveys (i.e. data collection strategies should not vary temporally and new sensors must be carefully calibrated to preexisting datasets). Consistency in photo collection (i.e. scheduling and no. of photo pairs) will enhance the comparison of temporal solutions (Gómez-Gutiérrez et al., 2014). Also, consider site visits at a particular time of day. (4) Process and generate photogrammetric solutions using the same software package and similar input processing parameters. And, a calibrated camera will always yield better solutions.

## 5 Conclusions

Comparative evaluations were completed using terrestrial LiDAR and photogrammetry, both terrestrial and aerial (UAV). None of these methods were without limitation and the ultimate goal of the data collection effort should guide the planning phase of the project. One cautionary note: without GCP there is no reasonable expectation that temporal activities will be successful. Although GCP may increase the workload during data acquisition, this is the only realization that will assure global alignment, minimize project error and enhance process theory development. While adherence to conventional ground methods for GCP establishment are essential for accurate temporal terrain characterization, the results presented herein are transferrable to larger survey areas with different terrain and surface characteristics. In terms of survey choice, all results point to a financial and temporal question. What is the project goal? What are the data expectations? Temporal assessment of gully channels and most geomorphic process descriptions can be accomplished with a camera and a few GCPs, whether on the ground or airborne. Each of the survey methods provided herein performed very well and, although the scoring was not very spectacular, the Fixed_61 data would be satisfactory for most static model evaluations. As expectations rise, so too will the planning and technology.

## Acknowledgements

This research was partially funded by grant 1359852 from the U.S. National Science Foundation. Authors would like to acknowledge the support provided by Nathan Stein (remote pilot for the quadcopter UAV), Daniel Murphy (remote pilot for the fixed-wing UAV), Justin Hobart, Tom Buman, Bob Buman, Sarah Anderson and Rick Cruse.

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

**Table 1: Datasets generated by three distinct surveying methods for the purpose of quantifying locational uncertainty in gully studies.**

| Dataset Identification | Dataset Description |
|---|---|
| Grd_2A | Channel left and right photo pair |
| Grd_2B | Upstream and downstream photo pair |
| Grd_4A | Channel left and right with corner left and right photo pair |
| Grd_4B | Upstream and downstream with corner left and right photo pair |
| Grd_4C | Upstream and downstream with channel left and right photo pair |
| Grd_6A | Upstream, downstream, channel left and right with corner left and right photo pair |
| Grd_8A | Upstream, downstream, channel left and right with both corner photo pairs |
| Quad_20m | Quadrotor flight at 20 m above ground surface |
| Quad_35m | Quadrotor flight at 35 m above ground surface |
| Fixed_61m | Fixed-wing flight at 61 m above ground surface |
| Fixed_122m | Fixed-wing flight at 122 m above ground surface |
| LiDAR | Terrestrial LiDAR survey (reference) |

*Dashed lines represent delineation between survey modes.

**Table 2: Metrics used in the ranking analysis of photogrammetric measurements.**

| Metric | Analysis | Description |
|---|---|---|
| M1 | Volume calculations from gridded data | Absolute volume difference between photogrammetric and LiDAR generated raster grids. |
| M2 | Volume calculations from gridded data | Iterative accumulation of individual raster grid cell elevation differences using absolute values. |
| M3 | Cross-section comparisons | Averaged coefficient of correlation between photogrammetric and LiDAR cross-section elevation values derived from raster grid analysis. |
| M4 | Cross-section comparisons | Averaged standard error between photogrammetric and LiDAR cross-section elevation values derived from raster grid analysis. |
| M5 | Cross-section comparisons | Averaged area percent difference between photogrammetric and LiDAR cross-section elevation values derived from raster grid analysis. |
| M6 | Spot elevation | Range of elevation difference between LiDAR three-dimensional irregular mesh and gridded photogrammetry. |
| M7 | Spot elevation | Variance of elevation difference between LiDAR and photogrammetry elevation values. |
| M8 | Spot elevation | Mean elevation difference between LiDAR and photogrammetry elevation values. |
| M9 | Spot elevation | Coefficient of correlation between LiDAR and photogrammetry elevation values. |
| M10 | Spot elevation | Standard error of linear regression between LiDAR and photogrammetry elevation values. |
| M11 | Normal to plane | Range of elevation difference between fitted plane to nearest neighbors LiDAR points and gridded photogrammetry. |
| M12 | Normal to plane | Variance of elevation difference between LiDAR and photogrammetry elevation values. |
| M13 | Normal to plane | Mean elevation difference between LiDAR and photogrammetry elevation values. |
| M14 | Normal to plane | Coefficient of correlation between LiDAR and photogrammetry elevation values. |
| M15 | Normal to plane | Standard error of linear regression between LiDAR and photogrammetry elevation values. |

**Table 3: Statistics comparing photogrammetry and terrestrial LiDAR using the point normal projected into fitted plane analysis. Residual values were calculated based on coordinate difference of all raster grid cells.**

| Dataset | Minimum Z axis difference (m) | Maximum Z axis difference (m) | Variance Z axis difference (m) | Mean Z axis difference (m) | Fitting linear model between photogrammetry and LiDAR in Z-axis | | | | |
|---|---|---|---|---|---|---|---|---|---|
| | | | | | Slope | Intercept | r-value | p-value | Standard error (m) |
| Fixed_122 | -0.0227 | 0.0231 | 0.0001 | -0.0038 | 0.8564 | 50.6434 | 0.9332 | 0.0000 | 0.0029 |
| Fixed_61 | -0.0229 | 0.0229 | 0.0001 | -0.0074 | 0.9284 | 25.2422 | 0.9793 | 0.0000 | 0.0010 |
| Quad_35 | -0.0205 | 0.0216 | 0.0000 | 0.0002 | 0.9935 | 2.2965 | 0.9959 | 0.0000 | 0.0004 |
| Quad_20 | -0.0148 | 0.0193 | 0.0000 | 0.0053 | 1.0315 | -11.1177 | 0.9970 | 0.0000 | 0.0004 |
| Ground_2A | -0.0226 | 0.0225 | 0.0000 | 0.0059 | 1.0239 | -8.4214 | 0.9937 | 0.0000 | 0.0005 |
| Ground_2B | -0.0192 | 0.0216 | 0.0000 | 0.0061 | 1.0258 | -9.0910 | 0.9985 | 0.0000 | 0.0003 |
| Ground_4A | -0.0219 | 0.0229 | 0.0000 | 0.0007 | 1.0027 | -0.9458 | 0.9949 | 0.0000 | 0.0005 |
| Ground_4B | -0.0212 | 0.0228 | 0.0000 | 0.0059 | 1.0203 | -7.1574 | 0.9984 | 0.0000 | 0.0003 |
| Ground_4C | -0.0218 | 0.0223 | 0.0000 | 0.0055 | 1.0178 | -6.2732 | 0.9976 | 0.0000 | 0.0003 |
| Ground_6A | -0.0204 | 0.0215 | 0.0000 | 0.0052 | 1.0219 | -7.7293 | 0.9984 | 0.0000 | 0.0003 |
| Ground_8A | -0.0197 | 0.0219 | 0.0000 | 0.0040 | 1.0177 | -6.2473 | 0.9982 | 0.0000 | 0.0003 |

| Dataset | Minimum Y axis difference (m) | Maximum Y axis difference (m) | Variance Y axis difference (m) | Mean Y axis difference (m) | Fitting linear model between photogrammetry and LiDAR in Y-axis | | | | |
|---|---|---|---|---|---|---|---|---|---|
| | | | | | Slope | Intercept | r-value | p-value | Standard error (m) |
| Fixed_122 | -0.0169 | 0.0164 | 0.0000 | 0.0004 | 1.0001 | -511.3216 | 1.0000 | 0.0000 | 0.0001 |
| Fixed_61 | -0.0180 | 0.0165 | 0.0000 | 0.0002 | 0.9974 | 12467.67 | 1.0000 | 0.0000 | 0.0000 |
| Quad_35 | -0.0138 | 0.0158 | 0.0000 | 0.0004 | 0.9998 | 917.4185 | 1.0000 | 0.0000 | 0.0000 |
| Quad_20 | -0.0112 | 0.0153 | 0.0000 | 0.0004 | 1.0016 | -7635.98 | 1.0000 | 0.0000 | 0.0000 |
| Ground_2A | -0.0161 | 0.0164 | 0.0000 | 0.0002 | 1.0020 | -9613.20 | 1.0000 | 0.0000 | 0.0000 |
| Ground_2B | -0.0166 | 0.0162 | 0.0000 | 0.0003 | 1.0025 | -11557.52 | 1.0000 | 0.0000 | 0.0000 |
| Ground_4A | -0.0165 | 0.0157 | 0.0000 | 0.0003 | 1.0004 | -1749.80 | 1.0000 | 0.0000 | 0.0000 |
| Ground_4B | -0.0165 | 0.0154 | 0.0000 | 0.0003 | 1.0023 | -10955.43 | 1.0000 | 0.0000 | 0.0000 |
| Ground_4C | -0.0165 | 0.0134 | 0.0000 | 0.0002 | 1.0021 | -9880.48 | 1.0000 | 0.0000 | 0.0000 |
| Ground_6A | -0.0163 | 0.0155 | 0.0000 | 0.0003 | 1.0021 | -9835.37 | 1.0000 | 0.0000 | 0.0000 |
| Ground_8A | -0.0172 | 0.0143 | 0.0000 | 0.0003 | 1.0016 | -7588.89 | 1.0000 | 0.0000 | 0.0000 |

| Dataset | Minimum X axis difference (m) | Maximum X axis difference (m) | Variance X axis difference (m) | Mean X axis difference (m) | Fitting linear model between photogrammetry and LiDAR in X-axis | | | | |
|---|---|---|---|---|---|---|---|---|---|
| | | | | | Slope | Intercept | r-value | p-value | Standard error (m) |
| Fixed_122 | -0.0111 | 0.0098 | 0.0000 | 0.0000 | 1.0002 | -88.1715 | 1.0000 | 0.0000 | 0.0001 |
| Fixed_61 | -0.0126 | 0.0094 | 0.0000 | -0.0001 | 1.0005 | -176.6591 | 1.0000 | 0.0000 | 0.0000 |
| Quad_35 | -0.0079 | 0.0042 | 0.0000 | -0.0001 | 0.9999 | 21.3139 | 1.0000 | 0.0000 | 0.0000 |
| Quad_20 | -0.0055 | 0.0055 | 0.0000 | -0.0001 | 0.9998 | 90.0612 | 1.0000 | 0.0000 | 0.0000 |
| Ground_2A | -0.0113 | 0.0065 | 0.0000 | -0.0001 | 0.9999 | 49.4894 | 1.0000 | 0.0000 | 0.0000 |
| Ground_2B | -0.0114 | 0.0048 | 0.0000 | 0.0000 | 0.9999 | 28.8352 | 1.0000 | 0.0000 | 0.0000 |
| Ground_4A | -0.0110 | 0.0061 | 0.0000 | 0.0000 | 1.0000 | -7.5965 | 1.0000 | 0.0000 | 0.0000 |
| Ground_4B | -0.0123 | 0.0047 | 0.0000 | 0.0000 | 0.9999 | 35.4773 | 1.0000 | 0.0000 | 0.0000 |
| Ground_4C | -0.0111 | 0.0052 | 0.0000 | 0.0000 | 0.9999 | 41.1742 | 1.0000 | 0.0000 | 0.0000 |
| Ground_6A | -0.0111 | 0.0037 | 0.0000 | 0.0000 | 0.9999 | 21.1864 | 1.0000 | 0.0000 | 0.0000 |
| Ground_8A | -0.0118 | 0.0031 | 0.0000 | 0.0000 | 1.0000 | 13.6678 | 1.0000 | 0.0000 | 0.0000 |

**Table 4: Statistics comparing photogrammetry and terrestrial LiDAR using the point vertical spot to TIN approach (Z-direction). Residual values were calculated based on the elevation difference of all raster grid cells.**

| Dataset | Minimum elevation difference (m) | Maximum elevation difference (m) | Variance elevation difference (m) | Mean elevation difference (m) | Fitting linear model between photogrammetry and LiDAR | | | | |
| --- | --- | --- | --- | --- | --- | --- | --- | --- | --- |
| | | | | | Slope | Intercept | r-value | p-value | Standard error (m) |
| Fixed_122 | -0.1388 | 0.0784 | 0.0019 | -0.0346 | 0.236 | 269.429 | 0.449 | <.0001 | 0.00212 |
| Fixed_61 | -0.0910 | 0.0506 | 0.0003 | -0.0144 | 0.849 | 53.133 | 0.939 | <.0001 | 0.00141 |
| Quad_35 | -0.0401 | 0.0538 | 0.0000 | 0.0001 | 0.987 | 4.589 | 0.992 | <.0001 | 0.00057 |
| Quad_20 | -0.0275 | 0.0438 | 0.0000 | 0.0058 | 1.029 | -10.133 | 0.994 | <.0001 | 0.00051 |
| Ground_2A | -0.0906 | 0.0659 | 0.0001 | 0.0066 | 1.023 | -8.144 | 0.985 | <.0001 | 0.00080 |
| Ground_2B | -0.0534 | 0.0535 | 0.0000 | 0.0070 | 1.024 | -8.505 | 0.996 | <.0001 | 0.00040 |
| Ground_4A | -0.0765 | 0.0723 | 0.0001 | 0.0001 | 0.985 | 5.428 | 0.986 | <.0001 | 0.00074 |
| Ground_4B | -0.0520 | 0.0509 | 0.0000 | 0.0066 | 1.019 | -6.531 | 0.996 | <.0001 | 0.00040 |
| Ground_4C | -0.0541 | 0.0517 | 0.0000 | 0.0061 | 1.014 | -4.810 | 0.994 | <.0001 | 0.00049 |
| Ground_6A | -0.0582 | 0.0506 | 0.0000 | 0.0059 | 1.020 | -7.193 | 0.996 | <.0001 | 0.00040 |
| Ground_8A | -0.0566 | 0.0486 | 0.0000 | 0.0045 | 1.016 | -5.689 | 0.996 | <.0001 | 0.00039 |
| Lidar | -0.0420 | 0.0394 | 0.0000 | -0.0001 | 1.001 | -0.257 | 0.998 | <.0001 | 0.00029 |

**Table 5: Simple statistics of comparative cross-section elevations generated using different surveying methods. Values were calculated for nine cross-sections individually and then averaged.**

| Dataset | Minimum elevation (m) | Maximum elevation (m) | Mean elevation (m) | Variance elevation (m) |
| --- | --- | --- | --- | --- |
| Fixed_122 | 352.551 | 352.628 | 352.596 | 0.0005 |
| Fixed_61 | 352.492 | 352.673 | 352.576 | 0.0019 |
| Quad_35 | 352.474 | 352.659 | 352.556 | 0.0025 |
| Quad_20 | 352.474 | 352.659 | 352.556 | 0.0025 |
| Ground_2A | 352.463 | 352.667 | 352.555 | 0.0025 |
| Ground_2B | 352.473 | 352.659 | 352.554 | 0.0024 |
| Ground_4A | 352.474 | 352.666 | 352.561 | 0.0022 |
| Ground_4B | 352.472 | 352.660 | 352.555 | 0.0024 |
| Ground_4C | 352.472 | 352.658 | 352.555 | 0.0024 |
| Ground_6A | 352.472 | 352.660 | 352.555 | 0.0024 |
| Ground_8A | 352.475 | 352.660 | 352.557 | 0.0024 |
| LiDAR | 352.472 | 352.660 | 352.555 | 0.0024 |

**Table 6: Volume difference between photogrammetry and terrestrial LiDAR raster grids generated from three-dimensional point clouds.**

| Dataset | Volume difference (m$^3$) | Cut volume (m$^3$) | Fill volume, (m$^3$) | Absolute volume difference* (m$^3$) |
|---|---|---|---|---|
| Fixed_122 | -0.0422 | 0.0078 | -0.0499 | 0.0577 |
| Fixed_61 | -0.0175 | 0.0021 | -0.0196 | 0.0217 |
| Quad_35 | 0.0002 | 0.0029 | -0.0027 | 0.0056 |
| Quad_20 | 0.0072 | 0.0079 | -0.0007 | 0.0085 |
| Ground_2A | 0.0081 | 0.0093 | -0.0012 | 0.0105 |
| Ground_2B | 0.0086 | 0.0088 | -0.0002 | 0.0090 |
| Ground_4A | 0.0003 | 0.0032 | -0.0029 | 0.0062 |
| Ground_4B | 0.0082 | 0.0084 | -0.0002 | 0.0086 |
| Ground_4C | 0.0076 | 0.0080 | -0.0005 | 0.0085 |
| Ground_6A | 0.0073 | 0.0076 | -0.0002 | 0.0078 |
| Ground_8A | 0.0056 | 0.0059 | -0.0003 | 0.0062 |

\* iterative accumulation of individual raster grid cell elevation differences using absolute values

**Table 7: Cross-section evaluation comparison between photogrammetry and LiDAR.**

| Dataset | Fitting linear model between photogrammetry and LiDAR | | | | | Mean area* percent difference | Minimum area* percent difference | Maximum area* percent difference |
| | Slope | Intercept | r-value | p-value | Standard error | | | |
|---|---|---|---|---|---|---|---|---|
| Fixed_122 | 0.311 | 242.937 | 0.618 | <0.0001 | 0.025 | 7.79% | 4.78% | 15.04% |
| Fixed_61 | 0.747 | 89.336 | 0.882 | <0.0001 | 0.025 | 3.18% | 1.77% | 4.44% |
| Quad_35 | 1.008 | -2.745 | 0.989 | <0.0001 | 0.009 | -0.05% | -0.23% | 0.43% |
| Quad_20 | 1.048 | -17.059 | 0.990 | <0.0001 | 0.009 | -1.36% | -1.70% | -0.64% |
| Ground_2A | 0.969 | 11.030 | 0.972 | <0.0001 | 0.015 | -1.57% | -2.49% | -0.90% |
| Ground_2B | 1.019 | -6.554 | 0.994 | <0.0001 | 0.007 | -1.63% | -1.84% | -1.43% |
| Ground_4A | 0.923 | 27.163 | 0.958 | <0.0001 | 0.018 | -0.03% | -0.74% | 0.69% |
| Ground_4B | 1.028 | -9.963 | 0.994 | <0.0001 | 0.007 | -1.55% | -1.71% | -1.38% |
| Ground_4C | 1.028 | -9.737 | 0.993 | <0.0001 | 0.008 | -1.44% | -1.61% | -1.24% |
| Ground_6A | 1.030 | -10.502 | 0.994 | <0.0001 | 0.007 | -1.38% | -1.53% | -1.08% |
| Ground_8A | 1.031 | -10.823 | 0.996 | <0.0001 | 0.006 | -1.06% | -1.26% | -0.76% |

\* Area calculations used a horizontal reference elevation of 353 meters.

**Table 8: Results from the ranking analysis based on difference metrics of multiple photogrammetric surveys applied to gully channel monitoring.**

| Volume and cross-section (M1-M5) | | Spot elevation (M6-M10) | | Normal to plane (M11-M15) | | Combined (M1-M15) | |
|---|---|---|---|---|---|---|---|
| Dataset | Points | Points | Dataset | Points | Dataset | Points | Dataset |
| Ground_8A | 49 | Ground_8A | 50 | Ground_2B | 44 | Ground_8A | 140 |
| Quad_35 | 42 | Ground_4B | 41 | Ground_6A | 42 | Ground_6A | 121 |
| Ground_6A | 38 | Ground_6A | 41 | Ground_8A | 41 | Ground_4B | 114 |
| Ground_4A | 37 | Quad_20 | 37 | Ground_4B | 41 | Quad_35 | 111 |
| Quad_20 | 35 | Quad_35 | 36 | Quad_20 | 35 | Ground_2B | 107 |
| Ground_4B | 32 | Ground_4C | 34 | Quad_35 | 33 | Quad_20 | 107 |
| Ground_4C | 32 | Ground_2B | 33 | Ground_4C | 31 | Ground_4C | 97 |
| Ground_2B | 30 | Ground_4A | 25 | Ground_4A | 26 | Ground_4A | 88 |
| Ground_2A | 20 | Ground_2A | 16 | Ground_2A | 15 | Ground_2A | 51 |
| Fixed_61 | 9 | Fixed_61 | 12 | Fixed_122 | 13 | Fixed_61 | 30 |
| Fixed_122 | 6 | Fixed_122 | 5 | Fixed_61 | 9 | Fixed_122 | 24 |

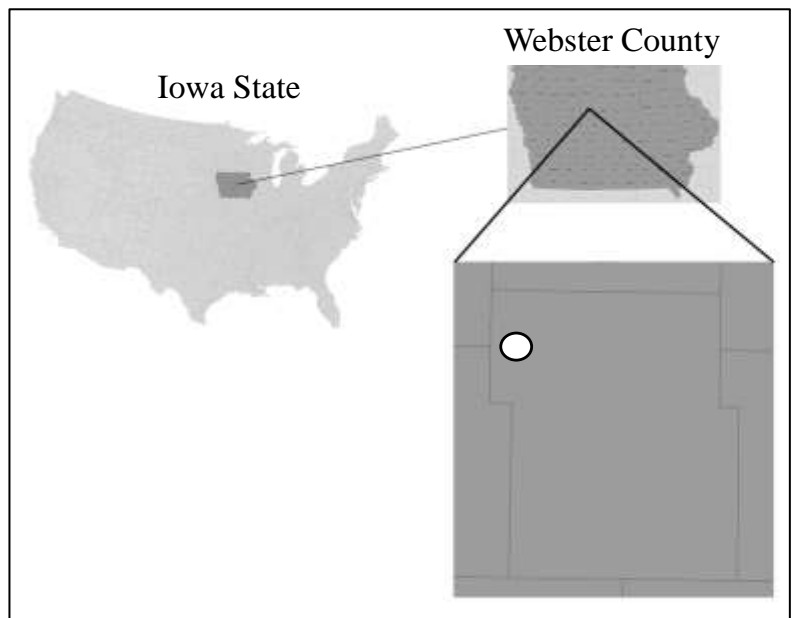

**Figure 1: Study site location used in the evaluation of close-range photogrammetric surveys of ephemeral gully channels.**

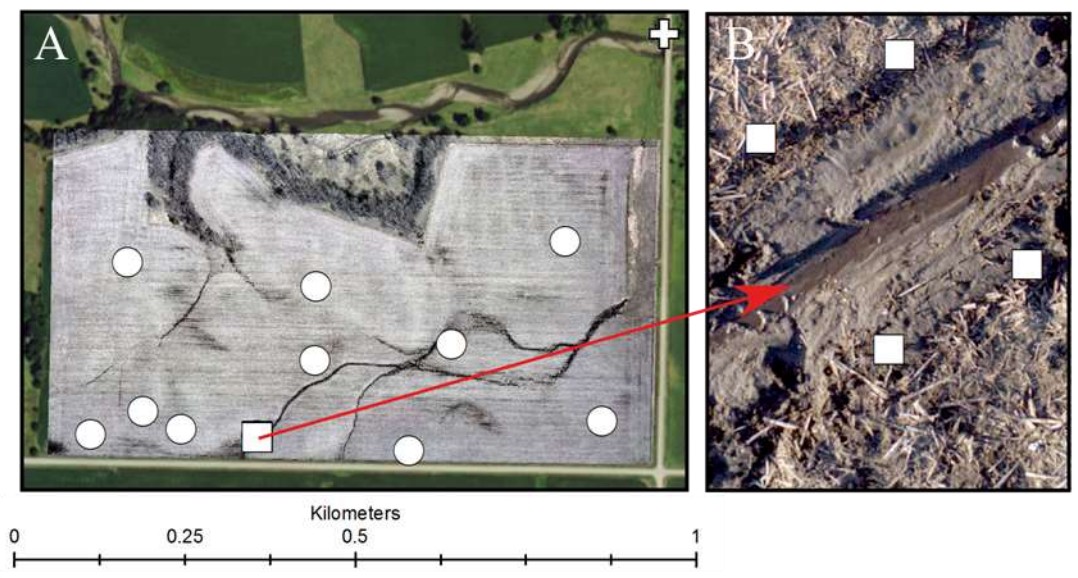

**Figure 2: (A) Study site with field ground control points (GCPs; circles) and state monument (cross) on bridge in upper right corner. (B) Selected AOI with channel GCPs (squares) for detailed surveys and comprehensive evaluation.**

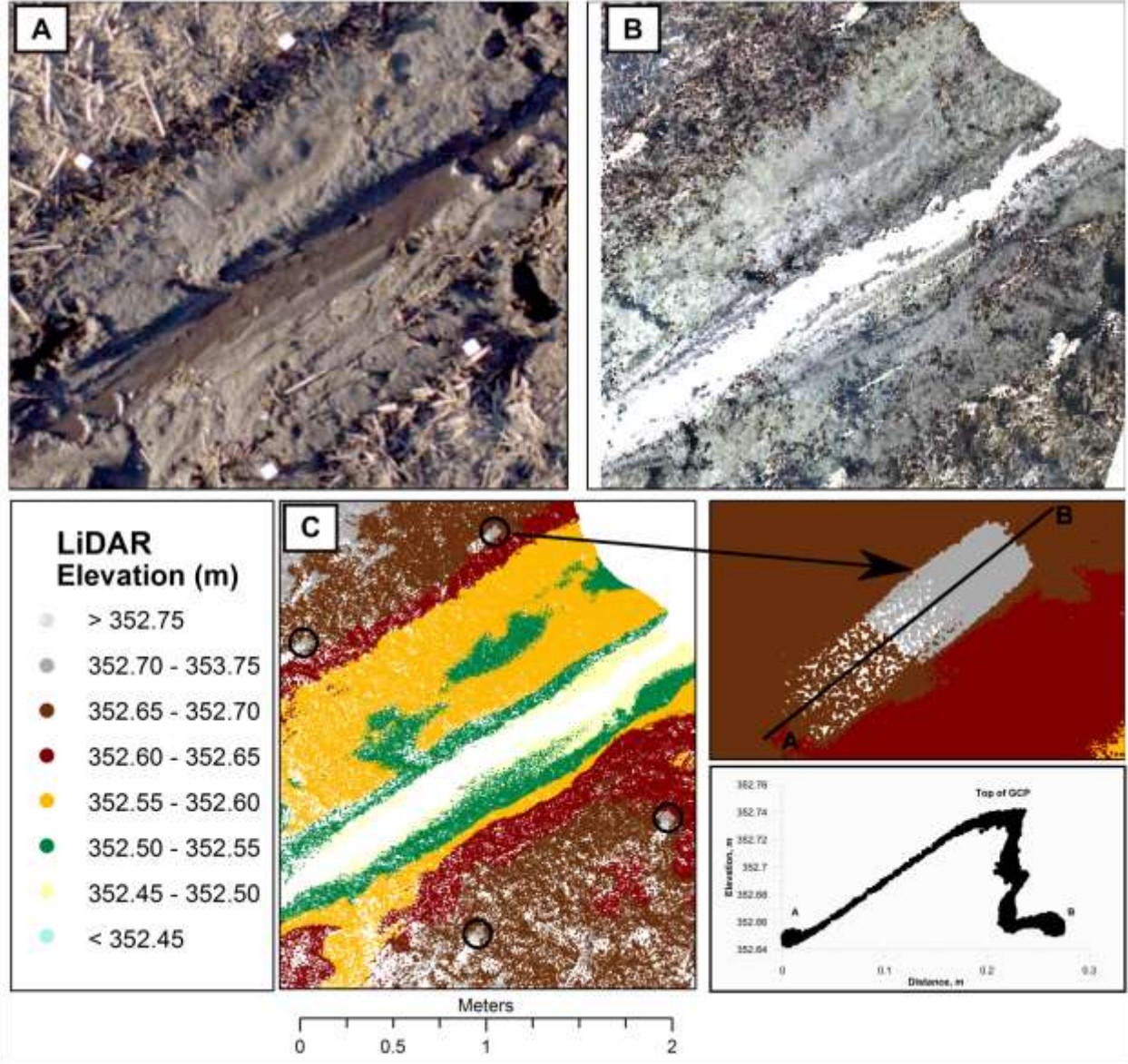

**Figure 3: Limitations of terrestrial LiDAR survey used herein as reference dataset. (A) Photograph of AOI shows water in the channel, which limits laser pulse return to the sensor causing sampling gaps in the point cloud (B), and presence of high relief features (GCPs) in the DEM (C) where sharp edges that cause the generation of multiple laser pulse returns due to split footprint effect.**

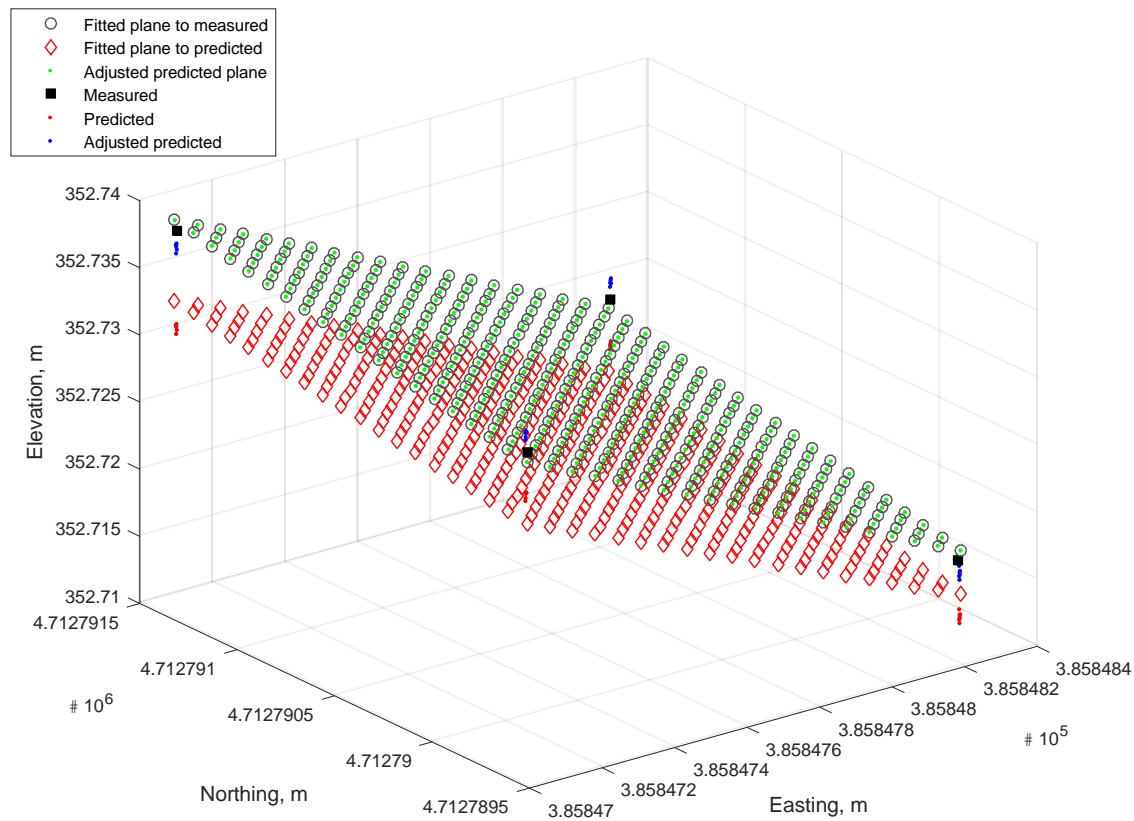

**Figure 4: Determination of Afine transformation matrices using Iterative Closest Point (ICP) methodology.**

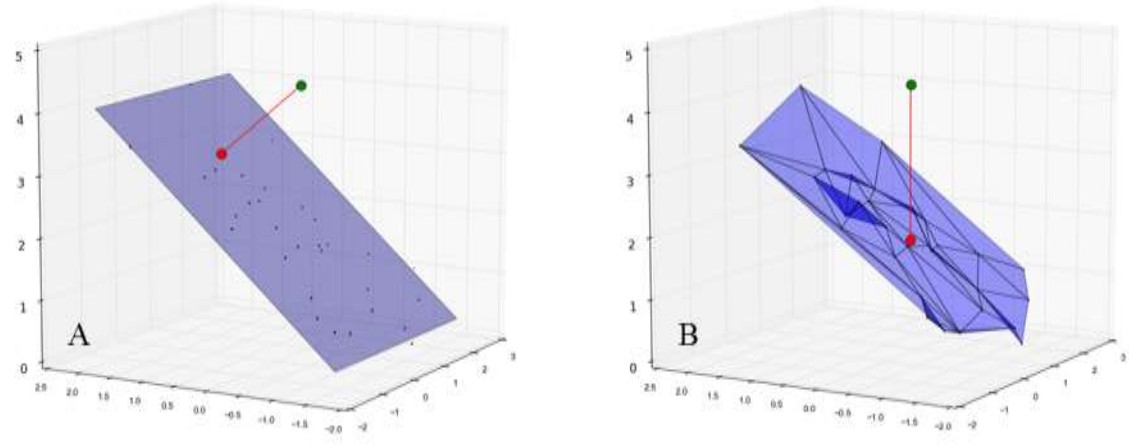

**Figure 5: Schematic representation of positional accuracy analysis. Black dots represent reference dataset (LiDAR) and green circle represents point being evaluated from photogrammetry. (A) Red circles represent the normal projection of green point onto the tangential plane fitted to the reference dataset and (B) the vertical projection of the green point into the three-dimensional reference triangular irregular network (TIN).**

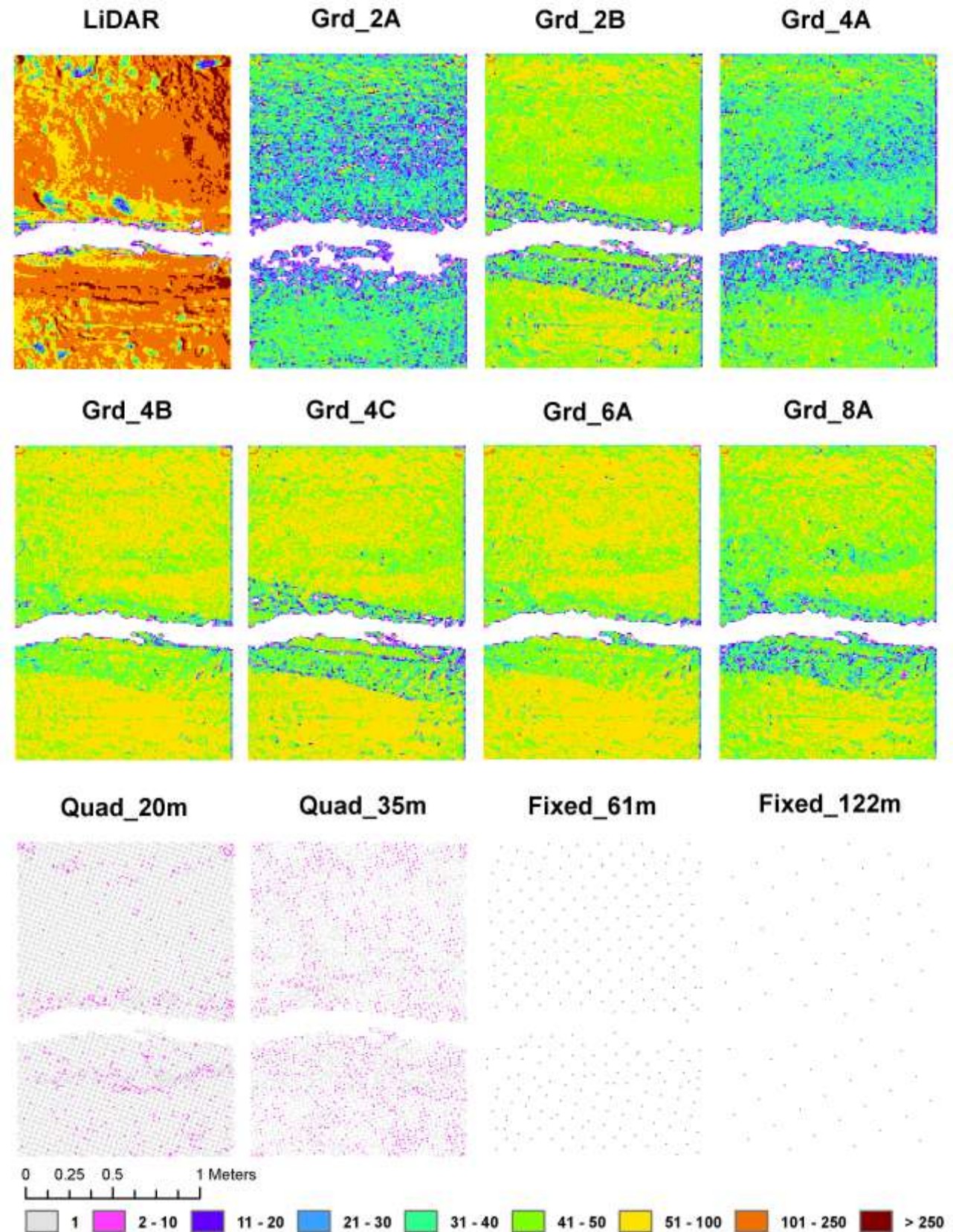

**Figure 6: Sampling intensity using the quadrat method for each dataset considered. Individual color represents point sampling count intervals within a 1x1cm virtual grid. Points located in the channel were removed to match the area covered by the LiDAR dataset (herein considered as reference).**

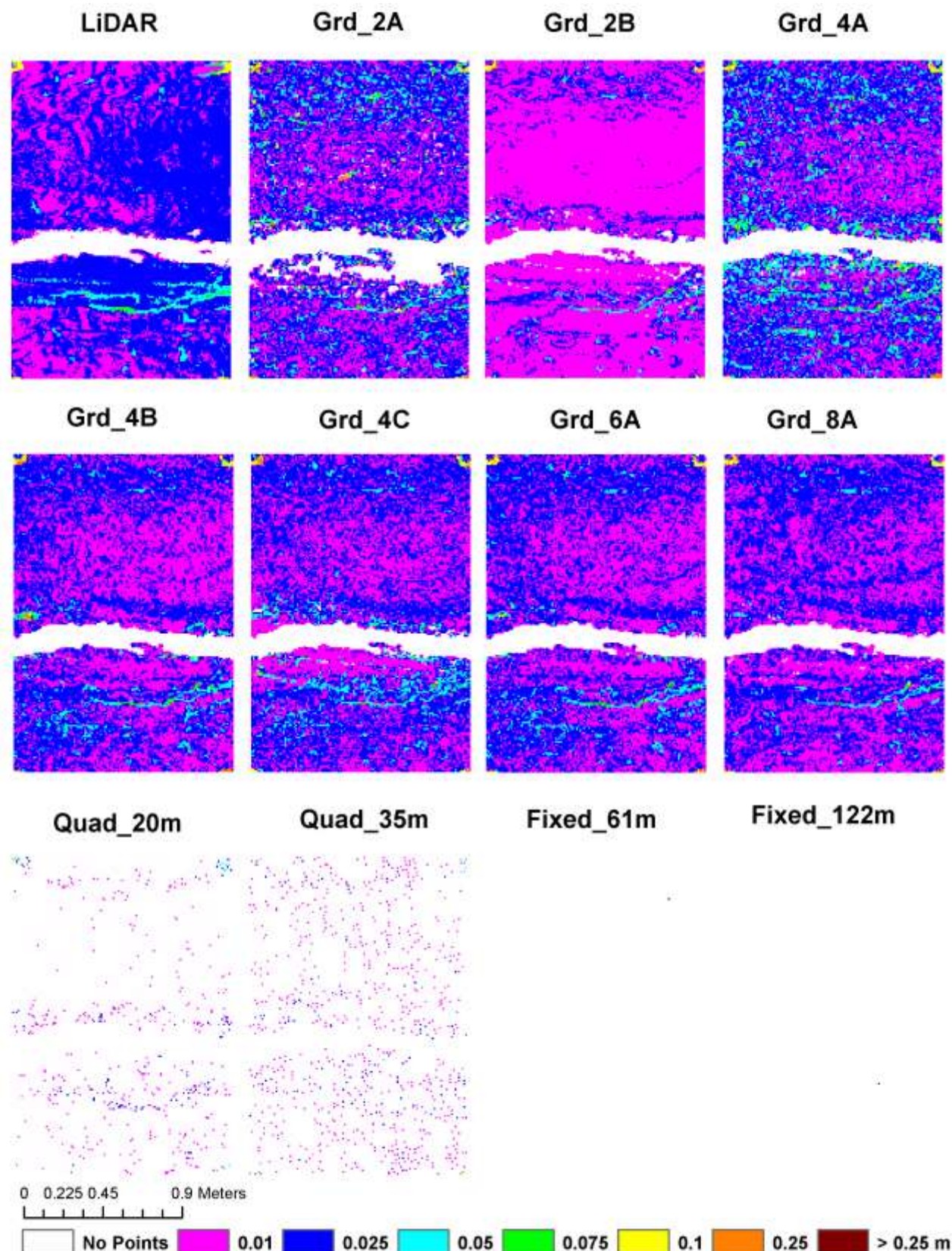

**Figure 7: Elevation range (difference between minimum and maximum elevation) represented as individual colors within a 1x1cm virtual grid. Points located in the channel were removed to match the area covered by the LiDAR dataset (herein considered as reference).**

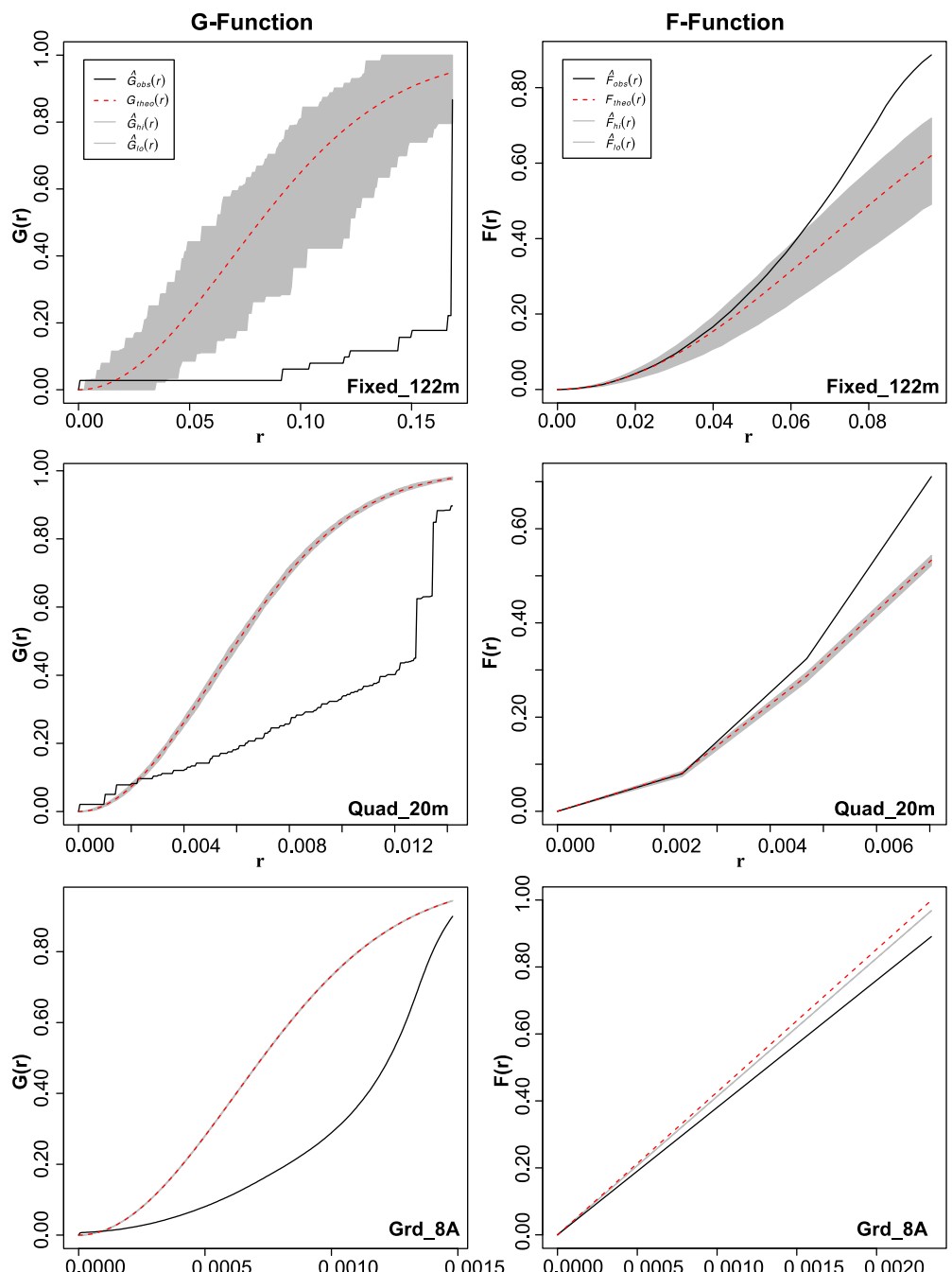

**Figure 8: Results of G-Function and F-Function analysis for Fixed_122m, Quad_20m, and Grd_8A datasets.**

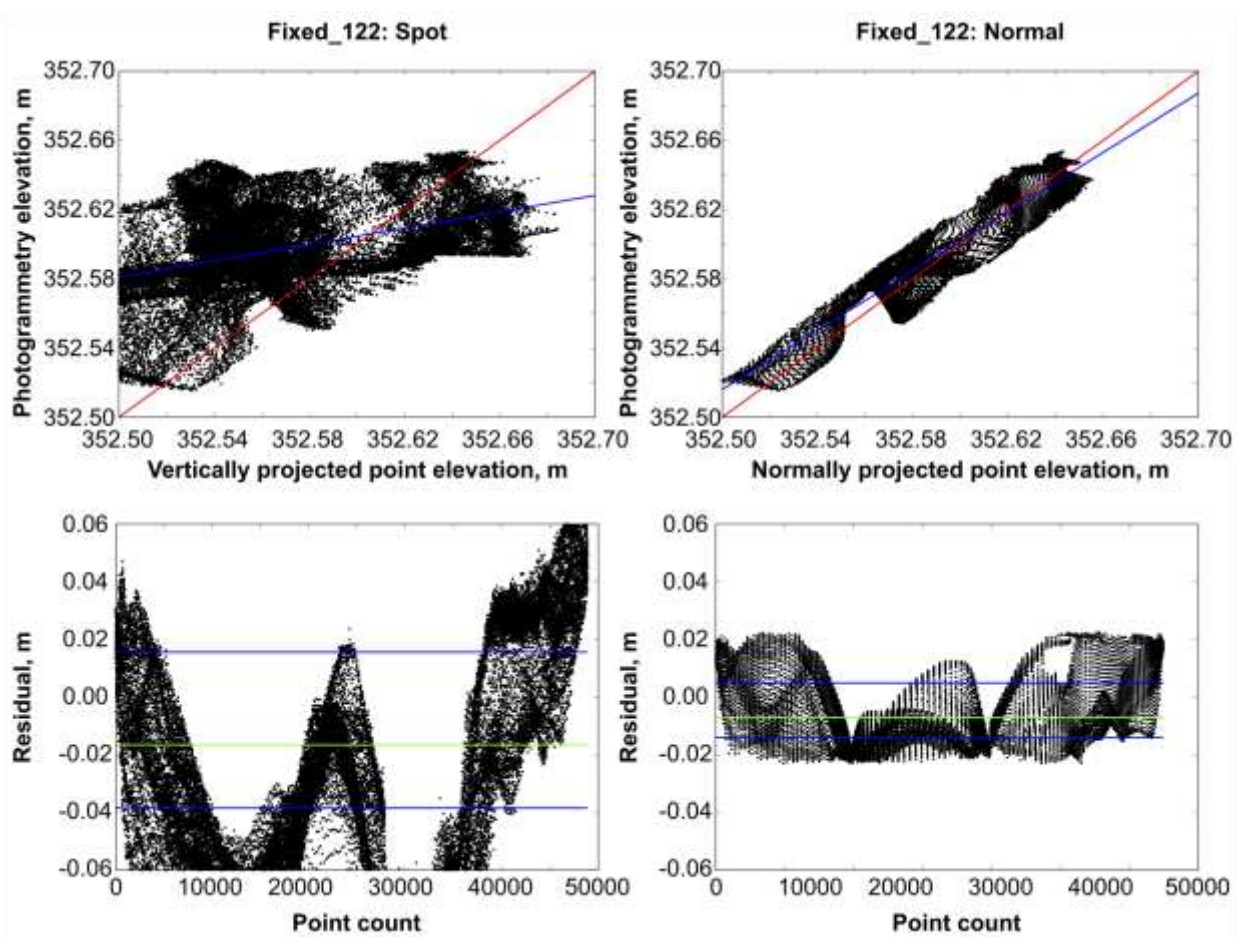

**Figure 9: Fixed-wing, at 122 m flight altitude, spot elevation comparison (left column) and normal to plane comparison (right column) of photogrammetry and LiDAR point cloud data with fitted line through the elevations (blue; Spot) and 1:1 line (red; Spot), along with the residuals (green is mean residual and blue lines are 25 and 75 percentiles).**

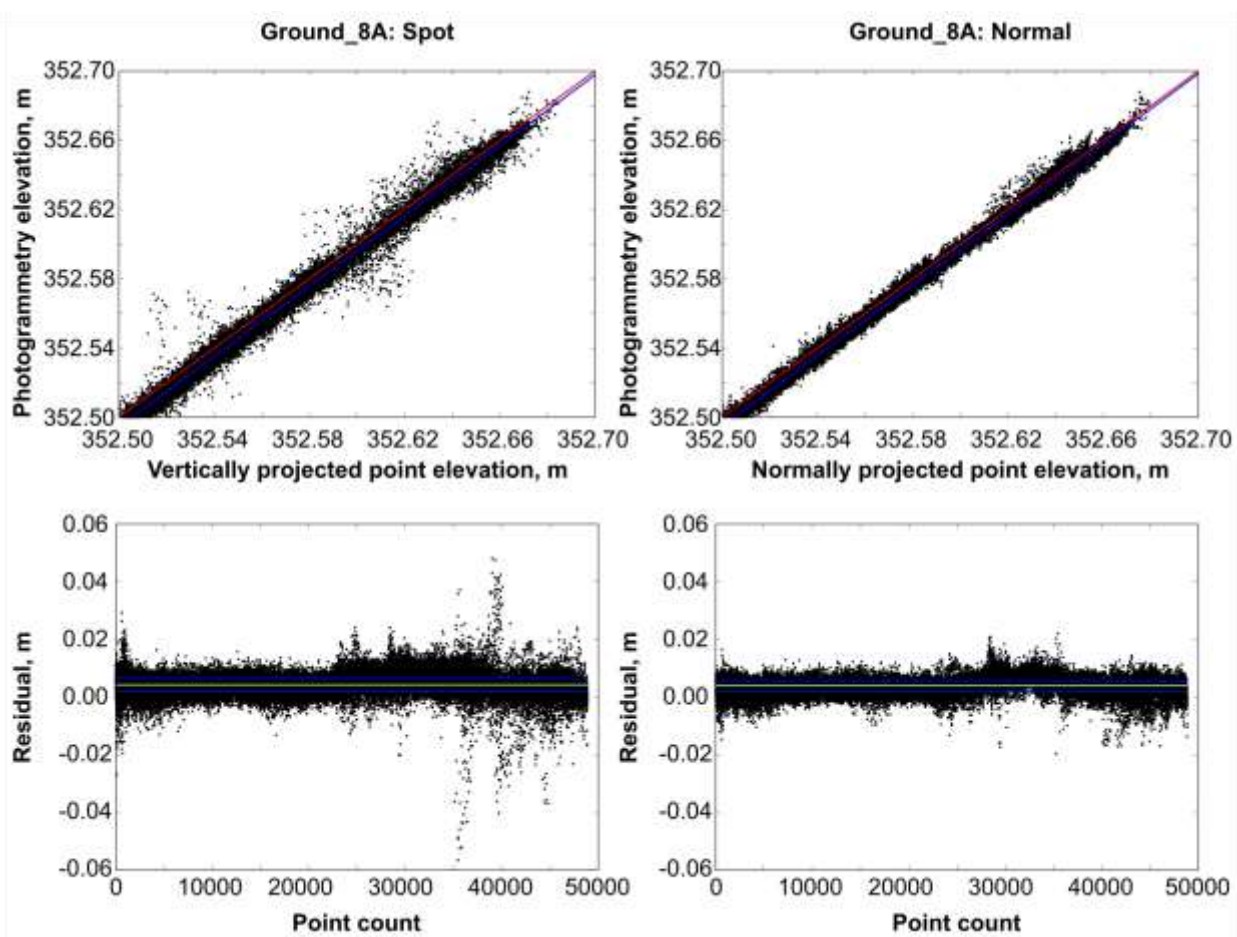

**Figure 10: Four photo pair (Ground_8A) spot elevation comparison (left column) and normal to plane comparison (right column) of photogrammetry and LiDAR point cloud data with fitted line through the elevations (blue; Spot) and 1:1 line (red; Spot), along with the residuals (green is mean residual and blue lines are 25 and 75 percentiles).**

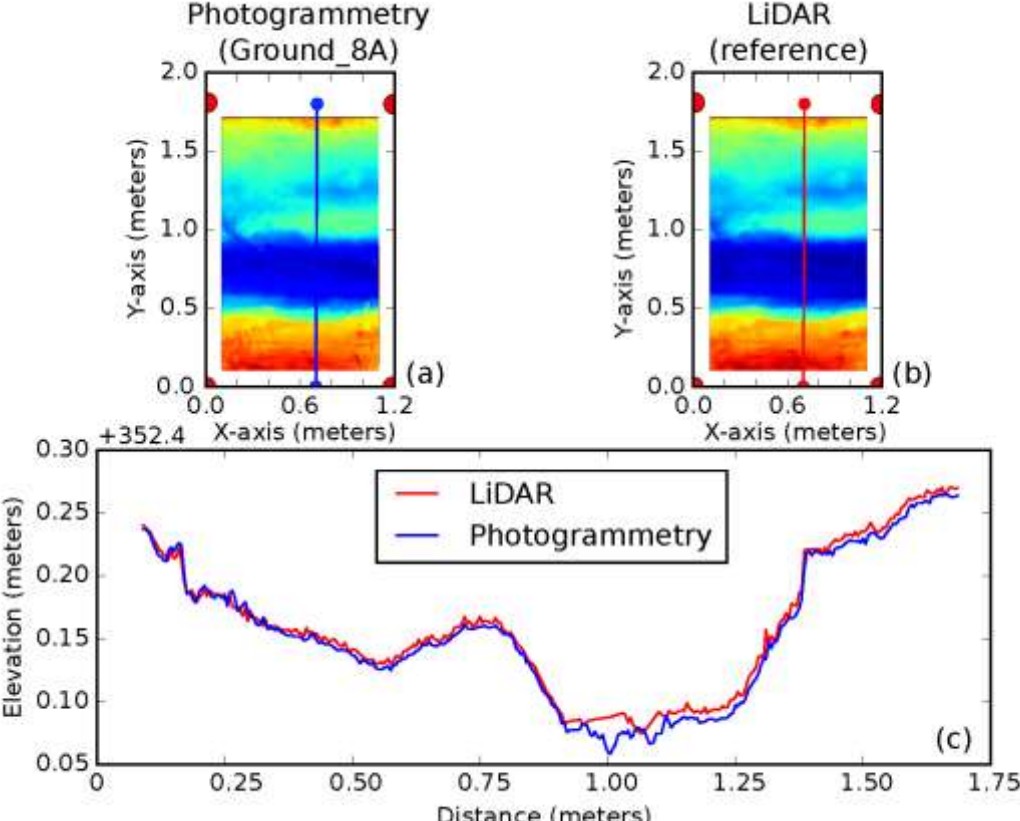

**Figure 11: Illustration of three-dimensional point cloud interpolation into raster grids for volume and cross-section analysis of gully monitoring and geomorphologic quantification. Direct comparison of Ground_8A photogrammetry (a) and LiDAR (b) raster grid data with a highlight of one specific cross section (c). The cross section can be realized anywhere within the scene.**

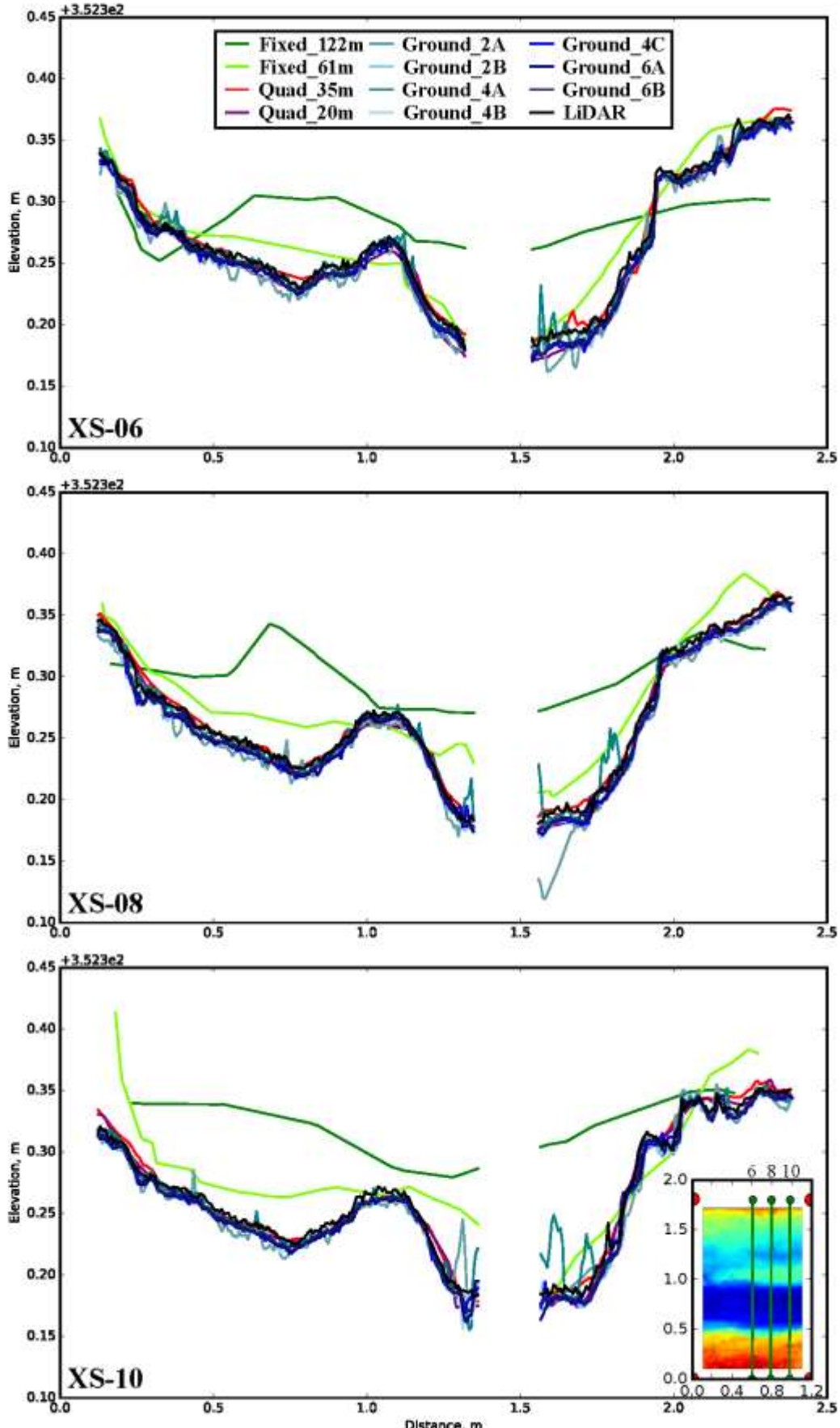

**Figure 12: Selected cross-sections generated from interpolating point-clouds into 5x5 mm raster grid file.**