# Peer review of "Quantifying uncertainty of high-resolution remotely sensed topographic surveys for ephemeral gully channel monitoring"

_Earth Surface Dynamics, 2017_

## Referee Comment (RC1) · Anonymous Referee #1 · 20 Feb 2017

The paper is interesting but in its present form, and based on the comments highlighted below, should be modified to improve the readability. The ideas could be interesting for the scientific communities, the methods and the assumptions resulted valid, but I suggest to improve some aspects of the analysis, the description of experiments and calculation. The reproduction of the experiment should be very difficult with the paper in this form. The conclusions resulted too general and no information repeatable and useful were provided. The authors should give proper credit to related work and the discussion (and conclusion) should clearly indicate their own new/original contribution. Therefore I suggest to improve the structure of the paper for a better reading, the discussion to compare the results obtained in the paper

[Figure]

**ESurfD**

Interactive
comment

to others recent paper, the conclusions have to be enlarged because in present form resulted very insignificant. I have highlighted many of these instances below in the specific comments. For the language, I suggest to verify the comma in the text. All the formulae, figures and tables have to be verified. I highlighted some corrections, but probably my suggestions are not complete. I suggest to improve the references considering some recent paper that compared different survey technology for erosion studies.

Please also note the supplement to this comment:
http://www.earth-surf-dynam-discuss.net/esurf-2017-3/esurf-2017-3-RC1-supplement.pdf

---

## Referee Comment (RC2) · 21 Feb 2017

REVIEW OF THE PAPER "Quantifying uncertainty of remotely sensed topographic surveys for ephemeral gully cannel monitoring" by Wells et al.

GENERAL COMMENTS This paper presents an interesting comparison of techniques-methods used to produce high-resolution topographic surfaces from pictures and laser. These techniques are used to reproduce the surface of an ephemeral gully. I think the results could be of interest for a broad audience, as they will increase the available datasets testing several methods to produce high-resolution topography. Before that, I suggest some modifications that could be used to improve the paper before considering its publication. I would say that these are minor revisions:

1) I miss some important references in the Introduction and Discussion sections, are there other papers comparing terrestrial or aerial SfM to LIDAR-LTS? This, in my opinion, may be addressed in the introduction section. Additionally, the result of these works could be used to enrich the discussion. Other important point that should be considered is DGPS. GCPs measured by a DGPS are ensuring, probably, a centimeter-level accuracy, so I suggest considering this point in the interpretation of the results. 2) In general, I consider that the text of the manuscript is well-organized but in some parts of the work there are too many sections and sub-sections, I suggest integrating some of them and this would increase the readability of the paper. 3) I also miss some methodological details, please see my comments below. The order of figures and tables should be reviewed. References should be reviewed (see my comments below).

SPECIFIC COMMENTS

Title: I understand that ephemeral gullies are, probably, not visible in satellite images but the terms "remotely sensed" could let to think in the use of this kind of info, so I recommend modifying the tittle to clarify this point, I suggest using "High-resolution remotely sensed". Abstract: Lack of standards in landform surveying is due, at least in part, to the variability, complexity, etc. of relief. Line 17, UAVs are not a technique more a platform, the technique is SfM or classical photogrammetry, so I suggest modifying this part of the abstract. In general for the text, when an acronyms is written, please, the first time when you use the extended form, use capital letters. INTRODUCTION L5: I guess you refer to Casali instead Casalli, by the way this reference is not in the reference list, please check. L11-14: I do not completely agree, LIDAR devices are now available to be the payload of a UAV so you can get great spatial-resolutions, on the other hand you can have the desired temporal resolution for LIDAR data, you just need money to pay for that, I suggest modifying the paragraph. L18: using multiple scan stations is not just for that, but more for avoiding shadows and normalizing the spatial resolution. L30: this paragraph does not flow with the rest of the introduction,

you start talking about point clouds but the reader could not know why you start discussing file formats, I suggest you start talking about the typical file format recorded by TLS, LIDAR and produced by SfM and later talking about the classical use of 2.5D file formats like DEM to represent surfaces. Equation 1: Does the ICP algorithm include the scale factor? I miss this in the explanation and formulation. Section 2.2.1 I miss error estimations for the DGPS, I guess 2-3 cm, would be nice to tell the readers about that. Section 2.2.4. I suggest including some points to support the selection of different software packages to run terrestrial and aerial photogrammetry. Section 2.2.4.1 I miss many details here: Overlaps, number of photos, UAV model, etc, etc. Section 2.4.1 Sampling intensity I have doubts here about the strategy used, did the authors sampled the number of points using a planimetric (XY) grid? In this case, they need to support this strategy. I think that estimations of volumetric point densities and 3D representations of this variable would work well in this case as. Problems using a grid in this case are important when representing vertical walls or headcut walls where points are distributed in the Z-coordinate direction, in this case, if you use a grid you will have a high value for the sampling intensity however, if you have a look to the point cloud, in some cases, you will realize that sampling intensity can be low.

Figure 10: better explanations with A)..., B)..., C)..., what kind of information was used? Figure 11: very difficult to understand and quite difficult to get conclusions from it! More than Fixed 122 is not working well for this Cross-section.

---

## Author Comment (AC1) · 27 Mar 2017

The paper is interesting but in its present form, and based on the comments highlighted below, should be modified to improve the readability. The ideas could be interesting for the scientific communities, the methods and the assumptions resulted valid, but I suggest to improve some aspects of the analysis, the description of experiments and calculation. The reproduction of the experiment should be very difficult with the paper in this form. The conclusions resulted too general and no information repeatable and useful were provided. The authors should give proper credit to related work and the discussion (and conclusion) should clearly indicate their own new/original contribution. Therefore I suggest to improve the structure of the paper for a better reading, the discussion to compare the results obtained in the paper to others recent paper, the conclusions have to be enlarged because in present form resulted very insignificant. I have highlighted many of these instances below in the specific comments. For the language, I suggest to verify the comma in the text. All the formulae, figures and tables have to be verified. I highlighted some corrections, but probably my suggestions are not complete. I suggest to improve the references considering some recent paper that compared different survey technology for erosion studies.

Specific Comments (obtained from RC1-supplement):
P1L25: I suggest to improve the introduction and the discussion with more recent studies. For example:
Di Stefano, C., Ferro, V., Palmeri, V., Pampalone, V., Agnello, F., (2017) Testing the use of an image-based technique to measure gully erosion at Sparacia experimental area. Hydrological Processes, 31 (3), pp. 573-585.
Hayas, A., Vanwalleghem, T., Laguna, A., Peña, A., Giráldez, J.V., (2017) Reconstructing long-term gully dynamics in Mediterranean agricultural areas. Hydrology and Earth System Sciences, 21 (1), pp. 235-249.
Smith M.W., Vericat D., (2015) From experimental plots to experimental landscapes: Topography, erosion and deposition in sub-humid badlands from Structure-from-Motion photogrammetry. Earth Surface Processes and Landforms, 40 (12): 1656-1671.
Vinci, A., Todisco, F., Brigante, R., Mannocchi, F., Radicioni, F., (2017) A Smartphone camera for the Structure from Motion reconstruction for measuring soil surface variations and soil loss due to erosion. Hydrology Research, in press
Vinci, A., Brigante, R., Todisco, F., Mannocchi, F., Radicioni, F., (2015) Measuring rill erosion by laser scanning. Catena, 124, pp. 97-108.
Vinci, A., Todisco, F., Mannocchi, F., (2016) Calibration of manual measurements of rills using Terrestrial Laser Scanning. Catena, 140, pp. 164-168.
Wheaton, J.M., Brasington, J., Darby, S.E. and Sear, D.A., (2010) Accounting for uncertainty in DEMs from repeat topographic surveys: improved sediment budgets. Earth Surf. Process. Landforms, 35: 136–156. doi:10.1002/esp.1886
Zheng, F., Xu, X., Qin, C., (2016) A Review of Gully Erosion Process Research. Nongye Jixie Xuebao/Transactions of the Chinese Society for Agricultural Machinery, 47.
We agree and have updated the manuscript, especially the introduction and discussion, to include some of the suggested references listed by the reviewer, as well as other pertinent sources.
Casalí, J., Loizu, J., Campo, M.A., De Santisteban, L.M., and Álvarez-Mozos, J.: Accuracy of methods for field assessment of rill and ephemeral gully erosion, Catena, 67(2): 128-138, 2006.
Casalí, J., Giménez, R., and Campo-Bescos, M.A.: Gully geometry: What are we measuring?, The Soil, 1: 509-513, 2015.
Di Stefano, C., Ferro, V., Palmeri, V., and Pampalone, V.: Testing the use of an image-based technique to measure gully erosion at Sparacia experimental area, Hydrol. Proc., 31: 573-585, doi: 10.1002/hyp.11048, 2017.

Eitel, J.U.H., Williams, C.J., Vierling, L.A., Al-Hamdan, O.Z., and Pierson, F.B.: Suitability of terrestrial laser scanning for studying surface roughness effects on concentrated flow erosion processes in rangelands, Catena, 87: 398-407, 2011.

Gómez-Gutiérrez, A., Schnabel, S. Berenguer-Sempere, F., Lavado-Contador, F., Rubio-Delgado, J. Using 3D photo-reconstruction methods to estimate gully headcut erosion, Catena, 120, 91-101, 2014.

Micheletti, N., Chandler, J.H., Lane, S.N.: Investigating the geomorphological potential of freely available and accessible structure-from-motion photogrammetry using a smart phone, Earth Surf. Process. Landforms, 40(4): 473–486, 2015.

Smith, M.W. and Vericat, D.: From experimental plots to experimental landscapes: topography, erosion and deposition in sub-humid badlands from structure-from-motion photogrammetry, 40: 1656-1671, 2015.

Vinci, A., Brgante, R., Todisco, F., Mannocchi, F., and Radicioni, F.: Measuring rill erosion by laser scanning, Catena, 124: 97-108. doi: 10.1016/j.catena.2014.09.003, 2015.

Wheaton, J.M., Brasington, J., Darby, S.E., and Sear, D.A.: Accounting for uncertainty in DEMs from repeat topographic surveys: improved sediment budgets, Earth Surf. Process. Landforms, 35: 136-156, 2010.

P2L11: Why? What's the problem?
With traditional airborne LiDAR, point collection is 12-30 points per square meter (depending on flight height). The mechanics of rill and ephemeral gully erosion require much finer resolution information. The DEM will be too smooth and will miss the position of the channel as well as morphology of the channel.

P2L40: at the same time? together?
During the same period. Each platform was run independent of the others at different times on the same day. (P3L17-22)
"Furthermore, measurements of the same site using identical methods and equipment, but performed at different time periods can also lead to three-dimensional registration errors. Therefore, the scope of this work was to evaluate multiple survey techniques and provide a framework for temporal studies of ephemeral gully channels. Three surveying platforms, with varying parameters, were independently evaluated for locational accuracy and applicability in generating information for model development/validation. The objectives of this study are twofold: to quantify the overall accuracy of the different survey configurations and to develop practical guidelines for the design and implementation of future ephemeral gully monitoring studies."

P3L3: could be very interesting, but where are?
The recommendations are outlined in Sec 4.2.
"Long-term photogrammetric monitoring of ephemeral gullies should be performed with systems and procedures that: (1) Provide a minimum sampling density to capture the overall and local terrain characteristics based upon study objectives (i.e. temporal headcut migration process understanding may require sub-centimeter resolution data, while temporal channel meander process understanding may only require decimeter resolution) (James and Robson, 2012; Gómez-Gutiérrez et al., 2014). The planning phase of the project must consider the physical characteristics of the process to be investigated, study site physical and environmental variables, and the available hardware and software. (2) Utilize static ground control points visible in comparable photo pairs in all time-step surveys (i.e. fixed known points within the scene provide checks to assure proper three-dimensional registration of temporal data) (e.g. Smith and Vericat, 2015). An organized scheme for control points must be realized for detailed multi-temporal quantitative assessment. Small variations in alignment within temporal surveys will introduce error into length, width, cross-sectional area and volume estimates (e.g. Casalí et al., 2015). Repeated realizations of GCP coordinates will always reduce error in survey solutions. (3) Collect the same number of photo pairs using the same sensor and with the same orientation in all time-step surveys (i.e. data collection strategies should not vary temporally and new sensors must be carefully calibrated to preexisting datasets). Consistency in photo collection (i.e. scheduling and no. of photo pairs) will enhance the comparison of temporal solutions (Gómez-Gutiérrez et al., 2014). Also, consider site visits at a particular time of day. (4) Process and generate photogrammetric solutions using the same software package and similar input processing parameters. And, a calibrated camera will always yield better solutions."

P3L27: Uppercase, verify in the text.
We agree and have made the correction.

P4L3: This technique was widely used, it is necessary this explanation? In the paper were used techniques less used and probably more interesting. I suggest a briefly description of the Terrestrial LIDAR. Probably the authors could be some photos of the equipment used for the paper.

The TLS was used as the "reference" and, this distinction required a more detailed description.

P6L23-24: where are the results of this analysis?

The results of this analysis and a more detailed description have been added to the manuscript. (P7L9-24; P9L8-20; Figure 8) Figure 8 is new to the revision.

"Two metrics were used to quantify the spatial pattern distribution: distances between events and between events and random points not in the pattern (void space). The G-Function, $G(r)$, defined as the cumulative frequency distribution of nearest-neighbor distances (Lloyd, 2010), provides the conditional probability that the distance between points (event-event) is less than the point distance threshold $(r)$. The empirical distribution is obtained for each distance $r$ by counting the number of points at distances less than or equal to $r$ from each point within the AOI. The theoretical distribution is obtained by assuming a completely random pattern with density ($\lambda$; estimated by the ratio of total number of points, divided by the area of the AOI), modeled as a Poisson process. Empirical values closer to the theoretical values indicate a random distribution, empirical values above the theoretical values indicate clustering, and empirical values below the theoretical values indicate a more regular distribution (Bivand et al., 2008). The F-Function, $F(r)$, defined as the cumulative frequency distribution of the distance to the nearest point in the AOI from random locations not represented within the AOI (Lloyd, 2010), provides the probability of observing at least one point (event) closer than $r$ to an arbitrary point within the AOI ("empty space" or "void" distances). Estimated and theoretical distributions are obtained similar to that of the G-function. The interpretation of graphed observed versus theoretical values indicates a regular pattern when the observed is above the theoretical values and clustering when it is below (Bivant et al., 2008). These point pattern analyses were performed using the Spatstat package in the R software package (Baddeley and Turner, 2005; Bivand et al., 2008).

Furthermore, points with the same X, Y and Z to the fifth decimal place were removed from the LiDAR dataset, indicating collection of redundancy information."

"Point pattern analysis was examined using the G- and F-functions (Figure 8). Each tests through an assumption of completely spatially random (homogeneous Poisson process), although interpretations for clustering and regularity are in opposition for each test (i.e. regular point spacing outcome for the G-function is below gray confidence bounds and for the F-function it is above). At first, the confidence bounds (gray envelope bounding the theoretical values (red dash line)) show that the Fixed_122 has a sparse point count. Looking at the G-function results, the data are clustered at distances of $0.02\ m$ (Fixed_122), $0.002\ m$ (Quad_20) and $0.001\ m$ (Grd_8A), then regularly distributed. This indicates that small distances occur less often than expected under the assumption of spatial randomness. The F-function results say that the data are randomly distributed to $0.06\ m$ (Fixed_122) and $0.025\ m$ (Quad_20), and clustered for Grd_8A (i.e. on short distances, fewer points are encountered than for a random pattern); although, the scale of $r$ should be acknowledged ($< 3\ mm$). All twelve datasets yielded observed G-function values below the theoretical values, indicating a regular sampling pattern. The terrestrial photogrammetric surveys did show slight clustering at small distances ($< 3mm$). Therefore, based on these metrics, a regular sampling pattern was observed, indicating that all locations within the study area were sampled with a similar sampling intensity (no areas where over sampled nor under-sampled) for all twelve datasets in this study."

[Figure]

**Figure 8: Results of G-Function and F-Function analysis for Fixed_122m, Quad_20m, and Grd_8A datasets.**

P7L20: Verify the units of the equations in all the paper.
The reviewer is correct. In equation 4 and 5, the elevation difference is multiplied by the cell area. Here, the cell area is 2.5E-5 square meters. The equations have been corrected. (P8L11-14)

P7L22: what is cs? It is cz?
As a continuance from the previous comment, the variable in question is $ca$, the cell area. The change has been made. (P8L13)

P8L5: In this table were reported the evaluation category, not the score.
The reviewer is correct. It was not intended to provide the scores within the materials section. Table 7 has been moved to Table 2 and provides a description of the metric tests as stated within the text of this section. Scores are provided in Table 8 within the results section.

P8L10: I suggest to better explain the influence of the sampling intensity. I suggest to insert a classification of "low", "medium", "high" sampling intensity. Therefore, in this paragraph should be reported the results of the point pattern analysis.
We agree and apologize for the confusion concerning the point pattern analysis. As mentioned in a previous comment, we enhanced the section to further describe the F- and G-functions. (P7L9-24; P9L8-20; Figure 8)  However, we do not share the reviewer's enthusiasm for ranking the intensity (i.e. low, medium and high).

P8L12: How? Probably a comment of a figure 11 or a better explanation could be useful.
Further discussion of the cross-section information is provided in Sec 3.4. (P10L33-P11L5) Figure 11 is now Figure 12.
"In the gridded elevation evaluations (Min, Max, Mean; Table 5; Figure 12), absolute difference from LiDAR was less than 0.02%, and these differences were only seen in the fixed-wing flights. The variance (i.e. roughness), however, does shoe that absolute differences from LiDAR were 131% (Fixed_122), 23% (Fixed_61), and 9% (Grround_4A). Comparing elevation information (Table 7; Figure 12) between photogrammetric cross-sections and LiDAR cross-sections through linear regression, indicates a coefficient of determination larger than 0.98 for all datasets, excluding the two fixed-wing flights. The standard error of this regression was less than 10 $mm$ for Quad_20, Quad_35, Ground_2B, Ground_4B, Ground_4C, Ground_6A, and Ground_8A. Following that, Ground_2A and Ground_4A had a standard error of approximately 17 $mm$ and the two fixed wings had a standard error of 25 $mm$. The average area percent difference for all cross-sections were within 1.5%, while the two fixed wings had 3% (Fixed_61) and 8% (Fixed_122). It is important to mention that the range of area percent difference is within ± 2%, while the fixed-wing systems had up to 15% percent difference. The error is huge for instance if this dataset was intended to be used for the purpose of development/calibration/validation of a soil erosion model."
The effect of sampling intensity can be explained as this: let us say you have one square meter and in one survey you collect 10 points and in another survey you collect a thousand points. If the surface of this square meter is a plane then both surveys will tell you the same thing; however, if the surface is rough, the survey with 10 points will not help you describe the terrain, as you may only collect peak or trough values, where the 1000 point survey will more effectively convey the characteristics of the terrain, the micro-topography (see papers by Haubrock et al., 2009 and Eitel et al., 2011 for discussions of surface roughness). So, if the surface is rough, low point counts will bias the volume estimates.
Eitel, J.U.H., Williams, C.J., Vierling, L.A., Al-Hamdan, O.Z., Pierson, F.B.: Suitability of terrestrial laser scanning for studying surface roughness effects on concentrated flow erosion processes in rangelands, Catena, 87: 398-407, 2011.
Haubrock, S.-N., Kuhnert, M., Chabrillat, S., Güntner, A., Kaufmann, H.: Spatiotemporal variations of soil surface roughness from in-situ laser scanning, Catena, 79: 128-139, 2009.

P8L27: What's mean? Clarify.
The two fixed-wing flights could not be adjusted by the methods described earlier within the text (see Sec. 2.3), since neither point cloud contained points on the channel GCPs. Therefore, the seemingly large deviation in elevation difference may be simply an alignment issue that we could not resolve/control or it is a bias we introduced in the raster process, since the results of the point analysis show that the fixed-wing data is clustered below 20 mm.

P10L3: I suggest to compare the results obtained with other recent papers.
We agree and have included appropriate references within the discussion for comparison.
Casalí, J., Giménez, R., and Campo-Bescos, M.A.: Gully geometry: What are we measuring?, The Soil, 1: 509-513, 2015.
Di Stefano, C., Ferro, V., Palmeri, V., and Pampalone, V.: Testing the use of an image-based technique to measure gully erosion at Sparacia experimental area, Hydrol. Proc., 31: 573-585, doi: 10.1002/hyp.11048, 2017.
Gómez-Gutiérrez, A., Schnabel, S. Berenguer-Sempere, F., Lavado-Contador, F., Rubio-Delgado, J. Using 3D photo-reconstruction methods to estimate gully headcut erosion, Catena, 120, 91-101, 2014.
Smith, M.W. and Vericat, D.: From experimental plots to experimental landscapes: topography, erosion and deposition in sub-humid badlands from structure-from-motion photogrammetry, 40: 1656-1671, 2015.

Wheaton, J.M., Brasington, J., Darby, S.E., and Sear, D.A.: Accounting for uncertainty in DEMs from repeat topographic surveys: improved sediment budgets, Earth Surf. Process. Landforms, 35: 136-156, 2010.

P10L20: Where is the Sec. 4.2.1?? For the point pattern analysis see the comments reported above.
Dyslexia. The reference should have been 2.4.1. We made the correction and enhanced the explanation for the point pattern analysis (P7L9-24; P9L8-20; Figure 8).

P11L1: I suggest to improve the guidelines for give more useful information.
The guidelines were intended to be broad, as many will not be interested in sub-centimeter characterizations. However, we did add some more information from our experiences to enhance the section. Because each individual will study a different problem, it is not possible to develop a cookbook that works for everyone. Instead, we described a path to minimize problems in processing and future temporal comparisons. We have added more details to our recommendations in this section. Our main message concerns GCPs and consistency. (P12L36-P13L12)
"Long-term photogrammetric monitoring of ephemeral gullies should be performed with systems and procedures that: (1) Provide a minimum sampling density to capture the overall and local terrain characteristics based upon study objectives (i.e. temporal headcut migration process understanding may require sub-centimeter resolution data, while temporal channel meander process understanding may only require decimeter resolution) (James and Robson, 2012; Gómez-Gutiérrez et al., 2014). The planning phase of the project must consider the physical characteristics of the process to be investigated, study site physical and environmental variables, and the available hardware and software. (2) Utilize static ground control points visible in comparable photo pairs in all time-step surveys (i.e. fixed known points within the scene provide checks to assure proper three-dimensional registration of temporal data) (e.g. Smith and Vericat, 2015). An organized scheme for control points must be realized for detailed multi-temporal quantitative assessment. Small variations in alignment within temporal surveys will introduce error into length, width, cross-sectional area and volume estimates (e.g. Casalí et al., 2015). Repeated realizations of GCP coordinates will always reduce error in survey solutions. (3) Collect the same number of photo pairs using the same sensor and with the same orientation in all time-step surveys (i.e. data collection strategies should not vary temporally and new sensors must be carefully calibrated to preexisting datasets). Consistency in photo collection (i.e. scheduling and no. of photo pairs) will enhance the comparison of temporal solutions (Gómez-Gutiérrez et al., 2014). Also, consider site visits at a particular time of day. (4) Process and generate photogrammetric solutions using the same software package and similar input processing parameters. And, a calibrated camera will always yield better solutions."

P11L16: See the comments reported above.
See responses reported above.

---

## Author Comment (AC2) · 27 Mar 2017

**A. G6mez-Gutlerrez (Referee):** alvgo@unex.es

REVIEW OF THE PAPER: "Quantifying uncertainty of remotely sensed topographic surveys for ephemeral gully cannel monitoring" by Wells et al.
GENERAL COMMENTS: This paper presents an interesting comparison of techniques- methods used to produce high-resolution topographic surfaces from pictures and laser. These techniques are used to reproduce the surface of an ephemeral gully. I think the results could be of interest for a broad audience, as they will increase the available datasets testing several methods to produce high-resolution topography. Before that, I suggest some modifications that could be used to improve the paper before considering its publication. I would say that these are minor revisions:

1) I miss some important references in the Introduction and Discussion sections, are there other papers comparing terrestrial or aerial SfM to LIDAR-LTS? This, in my opinion, may be addressed in the introduction section. Additionally, the result of these works could be used to enrich the discussion.
   We agree and have added several key references to enhance the Introduction and Discussion sections of the manuscript.
   Casalí, J., Loizu, J., Campo, M.A., De Santisteban, L.M., and Álvarez-Mozos, J.: Accuracy of methods for field assessment of rill and ephemeral gully erosion, Catena, 67(2): 128-138, 2006.
   Casalí, J., Giménez, R., and Campo-Bescos, M.A.: Gully geometry: What are we measuring?, The Soil, 1: 509-513, 2015.
   Di Stefano, C., Ferro, V., Palmeri, V., and Pampalone, V.: Testing the use of an image-based technique to measure gully erosion at Sparacia experimental area, Hydrol. Proc., 31: 573-585, doi: 10.1002/hyp.11048, 2017.
   Eitel, J.U.H., Williams, C.J., Vierling, L.A., Al-Hamdan, O.Z., and Pierson, F.B.: Suitability of terrestrial laser scanning for studying surface roughness effects on concentrated flow erosion processes in rangelands, Catena, 87: 398-407, 2011.
   Gómez-Gutiérrez, A., Schnabel, S. Berenguer-Sempere, F., Lavado-Contador, F., Rubio-Delgado, J. Using 3D photo-reconstruction methods to estimate gully headcut erosion, Catena, 120, 91-101, 2014.
   Micheletti, N., Chandler, J.H., Lane, S.N.: Investigating the geomorphological potential of freely available and accessible structure-from-motion photogrammetry using a smart phone, Earth Surf. Process. Landforms, 40(4): 473–486, 2015.
   Smith, M.W. and Vericat, D.: From experimental plots to experimental landscapes: topography, erosion and deposition in sub-humid badlands from structure-from-motion photogrammetry, 40: 1656-1671, 2015.
   Vinci, A., Brgante, R., Todisco, F., Mannocchi, F., and Radicioni, F.: Measuring rill erosion by laser scanning, Catena, 124: 97-108. doi: 10.1016/j.catena.2014.09.003, 2015.
   Wheaton, J.M., Brasington, J., Darby, S.E., and Sear, D.A.: Accounting for uncertainty in DEMs from repeat topographic surveys: improved sediment budgets, Earth Surf. Process. Landforms, 35: 136-156, 2010.

2) Other important point that should be considered is DGPS. GCPs measured by a DGPS are ensuring, probably, a centimeter- level accuracy, so I suggest considering this point in the interpretation of the results.
   We agree, however, the methods of GCP positioning and accuracy were not a primary focus of this manuscript. We report the error associated with the aligned datasets, assuming there are no ancillary errors in positioning. Here, we go to extreme lengths to ensure that the comparative datasets are in alignment and all comparisons stem from the locked channel GCP positions. We do agree that the error in GCP position should be reported and have added the error associated with static and kinematic collections for the unit we used in this study. (P4L12-16)

"All GCPs were surveyed using TopCon GR-3 dGPS survey equipment (TopCon Corporation, Tokyo, Japan; 10 mm horizontal and 15 mm vertical kinematic accuracy) to obtain relative position in reference to the state monument point. A static occupation (6 hrs; 3 mm horizontal and 5 mm vertical accuracy) was initiated with the base station, then all GCPs (field, channel and state monument) were surveyed with the rover (20 sec collection interval)."

3) In general, I consider that the text of the manuscript is well-organized but in some parts of the work there are too many sections and sub-sections, I suggest integrating some of them and this would increase the readability of the paper.
We agree and have integrated material to reduce the sub-sections. (P5L15-36)

4) I also miss some methodological details, please see my comments below. The order of figures and tables should be reviewed. References should be reviewed (see my comments below).
We have reordered the figures slightly and added flight details to address this issue. We reviewed and made corrections to the references.

SPECIFIC COMMENTS
Title: I understand that ephemeral gullies are, probably, not visible in satellite images but the terms "remotely sensed" could let to think in the use of this kind of info, so I recommend modifying the tittle to clarify this point, I suggest using "High-resolution remotely sensed".
We agree and have made the change.
"**Quantifying uncertainty of high-resolution remotely sensed topographic surveys for ephemeral gully channel monitoring**"

Abstract:
Lack of standards in landform surveying is due, at least in part, to the variability, complexity, etc. of relief.
We agree.

Line 17, UAVs are not a technique more a platform, the technique is SfM or classical photogrammetry, so I suggest modifying this part of the abstract.
We agree that UAV's are a platform for the technique of photogrammetry and do not see the confusion within these lines of text. "This work evaluated measurements of an ephemeral gully channel located in agricultural land using multiple independent survey techniques for locational accuracy and their applicability to generate information for model development/validation. Terrestrial and un-maned aerial vehicle-based photogrammetry platforms were compared to terrestrial LiDAR, defined herein as the reference dataset."

In general for the text, when an acronyms is written, please, the first time when you use the extended form, use capital letters.
We agree and corrected throughout the manuscript.

INTRODUCTION
L5: I guess you refer to Casali instead Casalli, by the way this reference is not in the reference list, please check.
Yes. We agree, corrected the text and added the reference.
Casalí, J., Loizu, J., Campo, M.A., De Santisteban, L.M., and Álvarez-Mozos, J.: Accuracy of methods for field assessment of rill and ephemeral gully erosion, Catena, 67(2): 128-138, 2006.

L11-14: I do not completely agree, LIDAR devices are now available to be the payload of a UAV so you can get great spatial-resolutions, on the other hand you can have the desired temporal resolution for LIDAR data, you just need money to pay for that, I suggest modifying the paragraph.
We agree and provided additional information concerning UAV LiDAR. However, that element was not within the purview of this research. Money will always be an issue at the edge of science. Even with this system, you would still need the information presented herein concerning the GCPs. (P2L18-24)
"Despite the large number of studies and methods developed to quantify positional errors in traditional airborne LiDAR surveys, this type of survey does not offer the temporal and spatial resolution necessary for quantitative monitoring of small-scale geomorphological characteristics (i.e. ephemeral gullies) necessary for process description; although, recent developments in UAV LiDAR systems provide 10mm survey-grade accuracy, one million

measurements per second, and 360° Field of View (FoV), all in <1.6 kg payloads. These UAV LiDAR systems can range from $100k to $400k (US dollar), dependent upon level of accuracy and data collection rate (see http://www.rieglusa.com as an example)."

L18: using multiple scan stations is not just for that, but more for avoiding shadows and normalizing the spatial resolution.
We agree and have added information to clarify the use of multiple scanning positions for LiDAR scene capture. (See P2L28)
"Overlapping the same area by multiple scans increases the overall sampling density and assists in occlusion and shadow avoidance, while normalizing spatial resolution."

L30: this paragraph does not flow with the rest of the introduction, you start talking about point clouds but the reader could not know why you start discussing file formats, I suggest you start talking about the typical file format recorded by TLS, LIDAR and produced by SfM and later talking about the classical use of 2.5D file formats like DEM to represent surfaces.
We agree and have added another paragraph to increase readability and help the flow. (P3L1-8)
"A particular point of interest is the general query posed by Wheaton et al. (2010) concerning real geomorphic change. With these evolving technologies, our ability to collect topographic information is seemingly limitless. At what point can we agree that the results describe "real" change over noise? The alignment of temporal-topographic elements is the most critical step when planning small-scale erosion studies (Smith and Vericat, 2015). Reliance on control points is the foundation of classical surveying. All surveys must close with a shot back to the initial occupation point. This too is the initiation of error propagation. A multitude of solutions exist for each set of photos and/or LiDAR points; however, the unique solution is bounded by the spatial and vertical positioning of the control points (Micheletti et al., 2015). Provided that alignment can be controlled, the next operation typically involves a culling process of some sort as the data shift into organized units."

Equation 1: Does the ICP algorithm include the scale factor? I miss this in the explanation and formulation.
No. The ICP was used to rotate and translate the point clouds only. We added this information to the sentence. (P6L16-20)
"Three-dimensional locational differences between the reference plane generated using the GPS survey (black squares in **Error! Reference source not found.**4) and the planes of each surveyed dataset (LiDAR and photogrammetry) were calculated using the Iterative Closest Point (ICP) algorithm (Besl and Mckey, 1994; James and Robson, 2012; Micheletti et al., 2015) implemented in MATLAB (MathWorks Inc., Natick, Massachusetts) and, since no scale issues were observed, no scaling factor was implemented in the ICP."

Section 2.2.1 I miss error estimations for the DGPS, I guess 2-3 cm, would be nice to tell the readers about that.
We agree and included the information as suggested. (P4L12-16)
"All GCPs were surveyed using TopCon GR-3 dGPS survey equipment (TopCon Corporation, Tokyo, Japan; 10 mm horizontal and 15 mm vertical kinematic accuracy) to obtain relative position in reference to the state monument point. A static occupation (6 hrs; 3 mm horizontal and 5 mm vertical accuracy) was initiated with the base station, then all GCPs (field, channel and state monument) were surveyed with the rover (20 sec collection interval)."

Section 2.2.4. I suggest including some points to support the selection of different software packages to run terrestrial and aerial photogrammetry.
We do not agree. There are far too many processing packages and there is no literature that places one product over another, so there is no need to support a processing software selection. Both are well cited within the literature and both have faults similar to other existing software packages.

Section 2.2.4.1 I miss many details here: Overlaps, number of photos, UAV model, etc., etc.
Of course. All flight parameters are now included in Section 2.2.4. (P5L16-26)
"Two UAV platforms were used to collect airborne photography (https://www.sensefly.com/home.html): fixed-wing (eBee) and quadrotor (albris). The fixed-wing platform had a 12 MP nadir camera (i.e. belly mount; Canon S110 RGB) and was deployed (eMotion2 v2.4.10) to capture the entire field boundary (Figure 2A) by throwing the craft in the air, then the craft flies, captures images and lands itself. Deployment software parameters were: Altitude (117 m, 58 m); Resolution (4.1 cm, 2 cm); Latitude Overlap (80%, 50%), Longitude Overlap (80%, 50%), Image Collection (356, 569), and Image Format (CR2(RAW)) for the two respective flights. The quadrotor platform had a 38 MP camera

mounted within an 180º vertical range head and was deployed (eMotionX v1.3.0) to capture both, the extent of the gully within the field and specific points of interest (i.e. gully cross sections; Gesch et al. 2015; Wells et al. 2016). The quadrotor was deployed through mission planning software. The craft takes off, flies and captures images, then lands itself. Deployment software parameters were: Altitude (35 m, 20 m); Resolution (0.7 cm, 0.5 cm), Latitude Overlap (75%, 75%), Longitude Overlap (80%, 80%), Image Collection (96, 146), and Image Format (dng(RAW)) for the two respective flights. During the flights, winds from the southeast ranged from 7 to 10 m/s and skies were clear."

Section 2.4.1 Sampling intensity I have doubts here about the strategy used, did the authors sampled the number of points using a planimetric (XY) grid?
Yes.

In this case, they need to support this strategy. I think that estimations of volumetric point densities and 3D representations of this variable would work well in this case as. Problems using a grid in this case are important when representing vertical walls or headcut walls where points are distributed in the Z-coordinate direction, in this case, if you use a grid you will have a high value for the sampling intensity; however, if you have a look to the point cloud, in some cases, you will realize that sampling intensity can be low.
We understand the point of contingency and agree with your assessment of 3D point distributions in the Z-coordinate direction; however, this inadequacy does not preclude the point sampling strategy using a planimetric grid. We submit that overhanging walls do not exist within the study channel examined here and further submit that all point counts were carried out prior to data gridding. In this study, the water film within the channel posed a problem, so we removed this area from consideration.

Figure 10: better explanations with A)..., B)..., C)..., what kind of information was used?
We regret the confusion. In an earlier version of the manuscript, we attempted to discuss the material as predicted (photogrammetry) and observed (LiDAR), and this figure made it through without correction. In the revision, we corrected the figure by stipulating the data within the panel (i.e. (a), (b) and (c)) as suggested. The figure represents a bit of scripting we use as a visualization tool. In the revision, this Figure 10 is Figure 11.

[Figure]

**Figure 11: Illustration of three-dimensional point cloud interpolation into raster grids for volume and cross-section analysis of gully monitoring and geomorphologic quantification. Direct comparison of Ground_8A photogrammetry (a) and LiDAR (b) raster grid data with a highlight of one specific cross section (c). The cross section can be realized anywhere within the scene.**

Figure 11: very difficult to understand and quite difficult to get conclusions from it. More than Fixed 122 is not working well for this Cross-section.

In the revision, this figure was moved to Figure 12. We regret the difficulty in understanding and have attempted to enrich the discussion to alleviate the confusion. First, we present these data to show that some of the platforms performed much better than others. And, we believe this point is obvious. Second, we agree with the reviewer that more than the Fixed_122 did not work well; however, we must determine the data usage before we consider whether the data is "good" or "bad". For instance, the Ground_4A data appears to have roughness elements in each of the cross-sections that place the data in the "bad" category for process investigation, while a fair amount of the metrics rank this dataset as being very close to the LiDAR. Overall, it ranks 8[th]. And, the Fixed_61 seems to be the other "bad" dataset; although the field scale survey and supplementary cross-section data may be appropriate for lumped model assessment (e.g. RUSLE2).

For example: (P10L33-P11L5)

"In the gridded elevation evaluations (Min, Max, Mean; Table 5; Figure 12), absolute difference from LiDAR was less than 0.02%, and these differences were only seen in the fixed-wing flights. The variance (i.e. roughness), however, does shoe that absolute differences from LiDAR were 131% (Fixed_122), 23% (Fixed_61), and 9% (Grround_4A). Comparing elevation information (Table 7; Figure 12) between photogrammetric cross-sections and LiDAR cross-sections through linear regression, indicates a coefficient of determination larger than 0.98 for all datasets, excluding the two fixed-wing flights. The standard error of this regression was less than 10 $mm$ for Quad_20, Quad_35, Ground_2B, Ground_4B, Ground_4C, Ground_6A, and Ground_8A. Following that, Ground_2A and Ground_4A had a standard error of approximately 17 $mm$ and the two fixed wings had a standard error of 25 $mm$. The average area percent difference for all cross-sections were within 1.5%, while the two fixed wings had 3% (Fixed_61) and 8% (Fixed_122). It is important to mention that the range of area percent difference is within ± 2%,

while the fixed-wing systems had up to 15% percent difference. The error is huge for instance if this dataset was intended to be used for the purpose of development/calibration/validation of a soil erosion model."

---

## Author Response (AR2)

**Revised Submission:**

**Associate Editor Decision: Reconsider after major revisions** (12 Apr 2017) by Anette Eltner

Comments to the Author:

Thank you very much for your interesting contribution to the Special Issue. The manuscript discusses an important issue and should be of relevance for a broader audience. The detailed analysis of accuracy performance of different data sources for very small scale geomorphological features from a large variation of observation distances is significant. Especially, the focus on GCPs used by different data sources for referencing and its significance for multi-temporal assessments of (i.e. low-magnitude) change detection is interesting. During the inter-active discussion process the authors explained their approach and answered all the comments by the referees in detail and made changes to the manuscript, where necessary.

Thank you. The comments and suggestions from reviewers were very thoughtful, appreciated and thoroughly addressed. As a result, an improved manuscript was produced. We appreciate the kind and positive comments from the AE.

However, there are two important aspects, which need further consideration. On the one hand, the assessment of one small excerpt within a large area captured via high resolution topography (HiRT) methods raises questions about the transferability of the results to other applications and the statistical foundation of the accuracy performance. For instance, how would different topographies and surface properties (e.g. texture) affect the performance of each method? This would also be relevant considering multi-temporal data acquisition.

The authors acknowledge variations in topography and surface properties will affect the performance of each method differently. However, the analysis of the small AOI is parallel and complementary to work on larger areas. The same attention to GCP is required for accurate terrain description, specifically when conducting long term studies at the same site, regardless of size.

In our work, terrain characteristic was fixed and the methods changed. The assessment of the small "excerpt" (i.e. AOI) provides detailed information on the performance character of the various capture techniques (i.e. LiDAR; terrestrial, non-SfM photogrammetry; aerial SfM photogrammetry), model assemblages (i.e. 2 to 8 photo pairs and SfM capture altitudes), and processing techniques. The approach allowed us to quantify differences between methods. The work is focused and does not mottle the character of the study by introducing multiple research topics.

There are a multitude of equipment and software to realize surface terrain and anyone, without any formal training, can, and do, go out and perform surveys of landforms, whether large or small in dimension. Many within our community attempt to use terrain information to inform prediction models or design algorithms of mechanistic change (i.e. headcut advance, channel widening). Therefore, realizing the limiting factors associated with a small AOI is, as indicated by the AE previously, "an important issue" and is transferrable to larger areas with different topographies and surface properties. We agree with the AE that a good follow up study would be to fix the survey method and vary topographic characteristics and solution model parameters.

With these statements in mind, we acknowledge that the AE raises a point that we should add within the text of the manuscript (see P13L23-25).

"While adherence to conventional ground methods for GCP establishment are essential for accurate temporal terrain characterization, the results presented herein are transferrable to larger survey areas with different terrain and surface characteristics."

Thus, the investigations will need extension to other locations, e.g. at the same large field that has been captured already. This should be highlighted in the manuscript.

We respectfully disagree. Rest assured that we routinely capture "other" locations; however, the topic of this manuscript does not require further evidence. At another time, we will use temporal data to discuss temporal issues, and at that time, we will refer back to this manuscript as a guide to assure the new reviewers that the temporal measurements that we utilize are aligned; therefore, conclusions concerning the temporal mechanics will be the focus of that new manuscript.

On the other hand, it has been mentioned that different settings of parameters and its influence at the final model were not assessed as they were beyond the scope of this study. However, this can result in significantly different surface models. How do you assure that your settings are the most suitable (also in direct comparison to the TLS data set)? This needs more consideration, experiments and explanation in the manuscript.

Varying input parameters when processing photo pairs in both standard and structure from motion photogrammetry software packages influence the quality of the three-dimensional point cloud generated. In both cases (standard and SfM), we used input parameters based on our prior experience in generating solutions with high sampling density AND small overall root mean square error values. The process was iterative. Performing a study of all possible input parameter configurations constitutes a daunting task (as hundreds of solutions can be generated). This is specifically true for the standard photogrammetric analysis (PhotoModeler Scanner), in which a large number of input parameters are available. Introducing this analysis into the current manuscript would distract the audience from the main objective and, a detailed analysis of varying software packages (PhotoModeler Scanner, Pix4D, Angisoft Photoscan, Erdas Imagine, etc) while exploring input parameter options constitutes an entirely new study.
As stated in the response to the previous comment, we respectfully disagree with the AE concerning "more experiments". Our experimental design considered the quantification of error when simulating multi-temporal capture of fixed topographic characteristics while, at the same time, varying capture techniques. Based on our results, we found that ground control is central for temporal studies.

Furthermore, there are some minor issues, described below, which need to be discussed.

Minor changes:

Accuracies regarding the SfM adjustment performance after GCP implementation are interesting information, especially because the authors perform a multitude of different SfM variations, which should be included in the study. Please, add these values.

There were four (4) SfM variations: Quad at two (2) elevations and Fixed-wing at two (2) elevations. To which accuracy do you refer? Adjustment performance after GCP implementation is listed in the tables. We also discuss the performance of each method.

In the abstract the authors state identical method and equipment can result in incorrect estimates of multi-temporal data due to registration error. If the registration approach is similar and assumed error of the GCPs nearly negligible, this cannot be the case. However, there are deviations, if measurement of GCPs (either in RGB images or intensity data of Lidar) is not exact (e.g. due to target fitting). Please, rephrase to a less rigid manner.

The sentence was rephrased to a less rigid manner.
"Measurements of the same site using identical methods and equipment, but performed at different time periods may lead to incorrect estimates of landform change as a result of three-dimensional registration errors."

In the introduction, please shorten line 12 to 21 at page 2 because the authors do not use airborne Lidar in their study and thus the reference to their study is not as obvious.

A large number of people have difficulty distinguishing between terrestrial and airborne LiDAR. When LiDAR is mentioned, their first thought is a small manned aircraft. Technology is changing rapidly and, if we cannot remember the lessons from our past then we are doomed to repeat. So, two lines within the introduction to clarify systems and similarity between positional errors does not seem unwarranted. And, part of this information was requested by one of the reviewers.

In chapter 2.2 please state the accuracy of your applied TLS device.

The instrument has a single point accuracy of 4 mm and surface accuracy of 2 mm at distances up to 150 m. The spot size is < 6 mm. Line 20 section 2.2.2 has been modified to include the requested information.

"The terrestrial LiDAR survey was conducted using a TopCon GLS 1500 (TopCon Corporation, Tokyo, Japan; $4\ mm$ single point and $2\ mm$ surface accuracy with a spot size $<\ 6\ mm$)."

Please, state the accuracy of each GCP after 20 seconds collection interval (chapter 2.2.1) because your work focuses on the impact of GCPs regarding (multi-temporal) registration aspects.

The accuracy of the GPS system was included in the manuscript. However, we do recognize and concede that the manufacturer's number and that realized during the survey were different by 1.0 mm. The correction applies to the base point only, as the solution obtained from the static session (OPUS) and that published by Iowa Department of Transportation were different in elevation by 6 mm, while the manufacturer of the GPS equipment used for collection stated an accuracy of 5 mm. Therefore, we have adjusted the information (P4L17-18).
In surveys of this type, all other collection points are adjusted, relative to the static solution. And, without similar static information on each GCP, we cannot speculate further than the information provided by the manufacturer (P4L14).

In chapter 2.2.2 at line 31 page 4, do the authors mean distance values (eventually leading to elevation values at smooth, plane surfaces) given the distance measurement accuracy (instead of vertical accuracy) because multiple reflections of the same surface originally result in this error due to the random measuring noise of the device? This can lead to height deviations but if the surface is more complex also positional errors are possible. Furthermore, please state how the 2 mm locational accuracy was measured?

The authors did not measure the TLS's location accuracy. Each laser pulse has a footprint (laser pulse is a cone). The reflection of the footprint depends on a number of parameters including the system's internal noise/accuracy. Here, we are specifically commenting on the noise inherent to the device with respect to height deviations as listed in the documentation of the laser scanner manufacturer. The next line should have stated "elevation variability" instead of "locational variability". We have corrected the line (P4L21).

"In this study, this elevation variability is estimated to be approximately $\pm 2\ mm$."

The ± 2 mm was intended to refer to the "fluff" or height variability within the LiDAR data.

In chapter 2.2.4 page 5 line 13 and line 28 please remove exterior because both put together - camera position and orientation - are referred to as exterior camera orientation.

The change was made (P5L30).

In chapter 2.3 please consider that the usage of points coinciding with GCPs is difficult in the case of SfM because in dependence of the GCP size image matching at the GCP location can be limited due to missing texture in the middle of these marked targets. Furthermore, in the approach of plane estimation the plane is "filled" with points for subsequent ICP. Please, add this information because else the mentioned point cloud could be misinterpreted with the actual ephermal gully representing data. Thus, furthermore scale becomes irrelevant during ICP because the points are created artificially as plane representation.

The lack of image texture (defined as reflectance variability) in features may limit the generation of points by automated photogrammetric software packages (a classical example is white sand). This was not the case in this study. The only datasets that did not have points within these targets were the Fixed_61 and Fixed_122 where the lack of points is attributable to low sampling intensity rather than lack of image texture at the site of the target.
The selection of points located within the four channel GCPs from each dataset was performed manually using two and three-dimensional visualization. The planes were fitted only to these selected points. The only exceptions were the two-fixed wing datasets where no points were located on the targets and, therefore, no locational adjustment was performed.
The idea of the ICP algorithm is to align two three-dimensional point clouds, one to another. This is performed through a minimization/optimization method that yields rotation, translation, and scaling matrices. Because we did not observe any scaling issue between all datasets, we applied the ICP algorithm to determine only rotation and translation matrices. The inclusion of the scaling comment was a response to a request from a reviewer.
We are confused by the "filled" comment, as we do not have this idea in the manuscript. The ICP was not used to "fill" the plane with points. The plane was defined by the channel GCPs and the rotation and translation matrices were defined by difference planes of the point cloud positions of the GCPs. The point clouds were adjusted by

rotation and translation, so all datasets could be analyzed from a common datum. Else, differences would also be effected by positional error. This is the main theme of the manuscript.

The comment of "artificially" created points is inaccurate. Measurements were made in the field for each method, separately. Solutions for each method were obtained using measured, and process corrected, control points. Deviations were found in the obtained solutions. All solutions were corrected to a common datum (channel GCPs) and those solutions were analyzed. We did not "fill" the plane, except for the fixed-wing flights with low sampling density, with "artificially" created points.

If these statements are in reference to the raster gridding process, this is business as usual within all science fields and many references exist that discuss both positive and negative aspects of raster gridding that are not germane here.

Page 7 line 3: Undersampling solely becomes relevant for positional accuracy of the interpolated model but not the points themselves. Please, rephrase.

A clarification was made by rewriting the sentence (P7L2).

Under sampled locations may be potential sources of error in quantifying geomorphologic change (i.e. cross-sectional areas, volumes, etc.), especially in surfaces with high relief.

The merit of the consideration of data scoring is questionable because there is no explanation to which extent different parameters are weighted or should be weighted. Please, be more specific to this regard.

All parameters were weighted equally. Weighting one parameter more than others depends on a full understanding of all factors and their influence on the overall outcome. At present, to the authors' knowledge, there is no accepted methodology within the scientific community. Our point with this manuscript is that we, as a group of concerned scientists, should come to an agreement on a systematic path forward concerning surveying collection methods and acceptable post-processing analysis techniques for terrain analysis. We are not proposing the data scoring presented herein as the final word, we just needed a simple way to understand the differences between the datasets. So, we improvised. We have no reason to weight one category over another. However, we do concede that we did not explicitly state that all variables were equally weighted and have added the text within Sec. 2.5 (P9L1).

"Simply, if a variable had a positive impact, it received a higher score and all variables were equally weighted."

In the results the authors refer to standard error but no explanation has been given. Please, add this information.

The standard error is the deviation of the observation from the true value. In all cases presented within the text, it is the deviation from a perfect linear fit to the LiDAR data. We believe this to be common knowledge concerning linear fit models and, similar to the other linear fit model parameters (i.e. $r^2$, $a$ and $b$), we do not see the need to define this parameter.

The evaluation of horizontal and vertical displacement reveals mainly alignment errors. Does this mean if this could be corrected or taken care of, deviations between the different methods become almost negligible? This would be an interesting result.

Please understand that your question is the focus of the manuscript. That is exactly what we have presented, with the deviations of the various methods. "Negligible" is a relative term, as it depends on the scale of the process you intend to study. In this manuscript, each method is viable for an array of studies. However, the fixed-wing flights at high altitude would not be a viable option for soil erosion *process* studies.

Results from this study document the importance of correct registration between surveys, specifically for temporal quantification of geomorphological changes. Based on our findings, even though all methods shared ground control points in data collection and processing, alignment to a common datum was only realized after utilization of ICP algorithm in the post-processing step. However, the results also show that, when assuming all the alignment errors are taken care of, there are differences in cross-sectional areas and volumes induced by varying sampling intensities, which are attributable to the method (i.e. terrestrial photo pairs, various UAV platforms) and, perhaps, solution parameters (Again, we point out that this point concerning solution parameter adjustment is an entirely new manuscript.).

However, because TLS accuracy has not been stated, yet, it is difficult to assess if the measured deviations are within the error domain of the scanning device, potentially neglecting any statements regarding the accuracy performance of all the tested methods. Furthermore do the results suggest that for ephermal gully observations solely close-range measurements are suitable?

We added the TLS accuracy information (P4L21).
No. Based on results from this study, assuming proper procedures were performed to minimize misalignment, the two quadrotors and even the Fixed_61 would be applicable. It all depends on the ultimate objective of the study.

In the discussion it is stated that GCP consideration cause great difficulties regarding multi-temporal assessment due to remaining alignment errors. Did the authors consider the GCP accuracy parameter, which is set during adjustment procedure? Was it close to zero, because else the models are not "lashed" to these positions during adjustment potentially resulting in different alignments for different data. Could the authors please be more specific?

All solutions contained GCP error on the order of mm. The models were "lashed" by the external geometry (GCPs); however, not all solutions contained the same GCP external geometry. For example, all terrestrial photogrammetry solutions utilized the channel GCPs, where the UAV flights utilized the field GCPs. This yielded very reasonable results but they were not in perfect alignment. In order to assess the difference in the methods, we had to remove the alignment bias (theme of this manuscript). We understand and share the frustration. (See rant in the discussion.)

In the conclusions, please moderate your statement regarding the necessity of GCPs for multi-temporal assessment. You refer to ephermal gullies but in other studies concentrating on change detection of larger magnitudes this might not apply.

We respectfully do not agree with either statement. There is no reason to change or modify the conclusions. GCPs are a necessity for multi-temporal assessments. This is not an opinion. The fact is, without GCP, any detection of change may be the result of misalignment. To ensure alignment, you must have control of the solution. Points that you assume do not change with time may move in time. Perhaps we can agree that each survey is unique and imposing a stringent methodology may not be the best solution for everyone; however, as shown within this manuscript, if we had assumed that we covered everything with good overlap and we could tie each survey together using matching points then the resulting differences between the survey methods would not be "truly" comparable, as they all had slightly different (~mm) positions for the core alignment points (i.e. channel GCPs). And, the reported differences would not have been change detection but misalignment. Think of our study as a temporal study in which we used different methods at different times and posted the change in the land surface over time. Although nothing moved, a few of the test cases would have presented large changes with time. With the GCP information, the solutions could be "lashed" in post-processing. Providing more accurate information. GCPs apply to all multi-temporal studies, within the scope of the targeted accuracy. As far as ephemeral gullies are concerned, the method of collection will need to suit the scope of the targeted accuracy. With model development or validation, close-range measurements will be a necessity.

Table 1 needs more specific explanation because in its current form the ground based SfM data acquisition scheme is not comprehensible. An additional figure might be helpful.

It is not clear to the authors what information is missing from Table 1. The authors did not receive any comments from reviewers having trouble with Table 1. Figure 2 presents the AOI with field and channel GCPs. Also, as noted in the table "Dashed lines represent delineation between survey modes (ground classical photogrammetry, airborne (UAV) structure from motion and ground-based LiDAR). The SfM data acquisition is only for the Quadrotor and Fixed-wing flights, these are described in detail within Sec. 2.2.4 and the flight elevations are provided in the table. Otherwise, photo positions are provided.

Table 4: What does the red line at the bottom (Lidar) refer to?

The statistics for the LiDAR raster grid in comparison to the LiDAR point cloud.

Please, add a legend to figure 3C because relief features are difficult to identify. Furthermore, the blue graphic does not have good quality. There can hardly anything be identified.

The figure caption contains the necessary information describing the effect highlighted in Figure 3C. "…and presence of high relief features (GCPs) in the DEM (C) where sharp edges that cause the generation of multiple laser pulse returns due to split footprint effect." Here, we point to a limitation that must be considered when analyzing LiDAR point cloud data. The line of points that are inserted into the dataset seem perfectly clear, at the top edge of the pin the scanner does not detect a surface until the shading from the occlusion is complete. For sure, you are aware that these points must be dealt with in a systematic manner, either due to point intensity or multi-occupation elimination. In either case, the point seems to be clear and neither of the reviewers raised a question concerning this material.

We have modified Figure 3C to clarify this issue.

Figure 9 and 10 need larger font sizes.

We have made the change.

[revised manuscript text omitted]